# Biochemical and structural characterization of an inositol pyrophosphate kinase from a giant virus

Guangning Zong [1], Yann Desfougères [2], Paloma Portela-Torres[2], Yong-Uk Kwon [3],
Adolfo Saiardi [2✉], Stephen B. Shears [1✉] & Huanchen Wang [1✉]

## Abstract

**Kinases that synthesize inositol phosphates (IPs) and pyrophosphates (PP-IPs) control numerous biological processes in eukaryotic cells. Herein, we extend this cellular signaling repertoire to viruses. We have biochemically and structurally characterized a minimalist inositol phosphate kinase (i.e., *Tv*IPK) encoded by *Terrestrivirus*, a nucleocytoplasmic large ("giant") DNA virus (NCLDV). We show that *Tv*IPK can synthesize inositol pyrophosphates from a range of *scyllo*- and *myo*-IPs, both in vitro and when expressed in yeast cells. We present multiple crystal structures of enzyme/substrate/nucleotide complexes with individual resolutions from 1.95 to 2.6 Å. We find a heart-shaped ligand binding pocket comprising an array of positively charged and flexible side chains, underlying the observed substrate diversity. A crucial arginine residue in a conserved "G-loop" orients the γ-phosphate of ATP to allow substrate pyrophosphorylation. We highlight additional conserved catalytic and architectural features in *Tv*IPK, and support their importance through site-directed mutagenesis. We propose that NCLDV inositol phosphate kinases may have assisted evolution of inositol pyrophosphate signaling, and we discuss the potential biogeochemical significance of *Tv*IPK in soil niches.**

**keywords** Giant Virus; Inositol Phosphate; Cell Signaling; Kinase; Phosphate Geochemistry
**Subject Categories** Microbiology, Virology & Host Pathogen Interaction; Post-translational Modifications & Proteolysis

## Introduction

The ubiquitous nature of eukaryotic cellular signaling through the catalytic activities of the kinases that synthesize inositol phosphates (IPs) and inositol pyrophosphates (PP-IPs) has been extensively reviewed (Kröber et al, 2022; Laha et al, 2021; Nguyen Trung et al, 2022; Shears and Wang, 2019). Three IP kinase families can be distinguished by their individual specific sequence signatures and

their shared tertiary structures: the 'inositol polyphosphate kinase' (IPK) family, which includes the $I(1,4,5)P_3$ 3-kinases, the inositol phosphate multikinase (IPMK) and inositol hexakisphosphate kinases (IP6Ks); the inositol pentakisphosphate kinase ($IP_5$ 2K); the two ATP-grasp kinases, inositol 1,3,4-trisphosphate 5/6-kinase (ITPK1) and the diphosphoinositol pentakisphosphate kinases (PPIP5Ks). Until recently, there were no reports of these kinases having been identified during the screening of non-eukaryotic genomes (Laha et al, 2021; Randall et al, 2020). That situation was recently changed by the sequencing of an archaea genome (*Lokiarcheum candidatus*) that unveiled an unexpected repertoire of eukaryotic signature proteins (Spang et al, 2015), including orthologs of ITPKs (Desfougeres et al, 2019). This development not only lead to the identification of an unexpected pathway of $IP_6$ synthesis in metazoans, but also uncovered a close association of IP metabolism with metabolic homeostasis (Desfougeres et al, 2019). The latter study testifies how valuable an evolutionary approach can be to furthering understanding of IP/PP-IP biology. This strategy is consolidated in the current study by the identification and characterization of IPK orthologs encoded by giant viruses, also known as nucleocytoplasmic large DNA viruses (NCLDVs) (Jansson and Wu, 2022).

The NCLDVs are abundant in both aquatic and soil niches (Jansson and Wu, 2022). Consequently, there is considerable biogeochemical significance to virus-mediated reprogramming of host cells into virocells (i.e., cells undergoing lytic infection), and the associated environmental release of many biological products (Moniruzzaman et al, 2020; Rigou et al, 2022; Schulz et al, 2020). The impact of NCLDVs upon soil ecology is drawing particular attention, especially in the context of the increasing rate of permafrost thawing due to climate change (Alempic et al, 2023; Jansson and Wu, 2022).

The founding member of the NCLDV family is *Acanthamoeba polyphaga mimivirus* (Raoult et al, 2004; Scola et al, 2003). Subsequently, the experimental model *Acanthamoeba castellanii* has been used as a host to isolate additional NCLDVs (Alempic et al, 2023; Colson et al, 2017); it is generally believed that amoeba are natural hosts for NCLDVs, including the mimiviruses (Colson et al, 2017). As an alternative to isolating virions, additional *mimiviridae* family members have been identified based on metagenome projects. One of these projects analyzed soil samples

[1]Inositol Signaling Group, Signal Transduction Laboratory, National Institute of Environmental Health Sciences, National Institutes of Health, Research Triangle Park, NC 27709, USA. [2]Medical Research Council Laboratory for Molecular Cell Biology, University College London, London, UK. [3]Department of Chemistry and Nanoscience, Ewha Womans University, 52, Ewhayeodae-gil, Seodaemun-gu, Seoul 03760, South Korea. ✉E-mail: a.saiardi@ucl.ac.uk; shears@niehs.nih.gov; huanchen.wang@nih.gov

taken from the Harvard Forest, from which the *Terrestrivirus* genome was characterized (Schulz et al, 2018); this encodes an IPK family member that we have now studied extensively. The biochemical and structural data that we have obtained lead us to discuss the potential nature and significance of *Tv*IPK to PP-IP turnover in soil niches. Additionally, our presentation of atomic-level structural details of PP-IP formation improves understanding of eukaryotic IP kinases. Finally, as the initial appearance of NCLDVs is proposed to predate eukaryotic origins (Guglielmini et al, 2019), we discuss the relevance of our results to ongoing debate on the evolution of PP-IP synthesis.

## Results

### Identification of *Terrestrivirus* inositol phosphate kinase *Tv*IPK

To identify divergent IP kinases we performed in silico genomic screening, using hidden Markov models (HMMs) as previously described (Laha et al, 2021), but including viral databases. This analysis was originally performed in January 2020; no additional viral information was retrieved when the screening was repeated in January 2023.

We identified three NCLDV genomes, all within the *Mimiviridae* family, that each encoded putative proteins hosting two important and conserved motifs found in all IPK family members (Appendix Fig. S1), i.e., D[I/V]K[I/L]G (the Lys residue stabilizes a catalytic transition state) and [V/I]DF (the Asp within this motif coordinates a catalytically essential Mg, and the Phe is a key component of a hydrophobic architectural spine) (Shears and Wang, 2019). We prepared a recombinant version of the *Terrestrivirus* protein, which we designate as *Tv*IPK.

### *Tv*IPK is a versatile inositol pyrophosphate kinase

The IPK family includes IP6Ks. The latter class of enzymes utilize ATP to add a 5-β-phosphate to *myo*-IP$_6$ (Fig. 1A) to form *myo*-5PP-I(1,2,3,4,6)P$_5$ (this PP-IP is often annotated as 5-IP$_7$) (Saiardi et al, 1999). We therefore incubated *Tv*IPK with 50 µM *myo*-IP$_6$ and 50 µM [γ-$^{33}$P]-ATP; HPLC analysis confirmed the accumulation of [$^{33}$P]-PP-IP with the elution characteristics of an IP$_7$ (Fig. 1B). Thus, *Tv*IPK is functionally related to the IP6K family.

IP6Ks also actively phosphorylate *myo*-(2OH)IP$_5$ (Saiardi et al, 2000a). We found that *myo*-(1OH)IP$_5$, *myo*-(2OH)IP$_5$ and *myo*-(3OH)IP$_5$ are all readily phosphorylated by *Tv*IPK (Fig. 1A–E; Appendix Fig. S2); in contrast, *Tv*IPK only weakly phosphorylated IP$_5$ isomers that possessed a lone OH group at the 4-, 5-, or 6-positions (Appendix Fig. S2). Interestingly, *Tv*IPK can also phosphorylate *myo*-I(1,4,5)P$_3$ and each of the *myo*-IP$_4$ isomers that are included in the canonical eukaryotic pathway to IP$_6$ synthesis (Appendix Fig. S2). Note that the addition of diphosphoinositol polyphosphate phosphohydrolase 1 (DIPP1) into the above incubations almost completely prevented any accumulation of phosphorylated product (Fig. 1F). The β-phosphate specific hydrolysis of a diphosphate group by DIPP1 verifies that *Tv*IPK phosphorylates a pre-existing phosphate on the IP$_5$ isomers, and not any of the available OH groups (e.g., see Zong et al, 2022).

These HPLC analyses additionally revealed *Tv*IPK-mediated accumulation of [$^{33}$P]-Pi, indicative of a non-productive ATPase (Fig. 1B–E, Table 2), as was recently described for human IP6K1, although in the latter case the activity was quite minor (<4% of kinase activity (Mohanrao et al, 2021)). The ratios of activities of phosphatase/kinase for *Tv*IPK varied with the nature of the substrate (Table 1) but are generally much greater than the corresponding ratios for IP6K1 (Mohanrao et al, 2021).

The *Terrestrivirus* genome was originally characterized from samples of forest soil (Schulz et al, 2018), in which potential host microorganisms are in abundance, including many amoeba (Denet et al, 2017). From among such protists IP metabolism has been best characterized in the social amoeba *Dictyostelium discoideum* in which, as in all eukaryotic cells, *myo*-IP$_6$ is particularly abundant (Stephens et al, 1990). Thus, an IPK synthesized by any soil-based virus can be expected to access *myo*-IP$_6$ from within its host. Additionally, soils contain *myo*-IP$_6$, *scyllo*-IP$_6$, *scyllo*-IP$_5$ and *scyllo*-I(1,2,3,4)P$_4$ (Fig. 1A), although it has not been determined if the latter is the L- and/or D-stereoisomer (Reusser et al, 2020; Turner et al, 2002). Each of these polyphosphates could be accumulated by foraging microorganisms through phagocytosis (Denet et al, 2017). We have found that *Tv*IPK actively phosphorylates both *scyllo*-IP$_6$ and *scyllo*-IP$_5$ while being stereoselective for the L-enantiomer of *scyllo*-I(1,2,3,4)P$_4$ (Fig. 1F, Appendix Fig. S2). The phosphorylation of five of the favored *scyllo*-IP substrates was also analyzed in the DIPP1-coupled assay, verifying that *scyllo*-PP-IPs are formed (Fig. 1F).

Substrate saturation plots indicate K$_M$ values of 81 µM for *myo*-IP$_6$ (Fig. 1G) and 1.6 mM for ATP (Fig. 1H). The unusually low affinity towards ATP compared to most other kinases is common to the canonical IP6Ks, and is considered an evolutionary adaptation that links IP$_7$ synthesis to cellular bioenergetic status (i.e., ATP availability) (Shears, 2018).

*Tv*IPK also phosphorylates *myo*-[$^3$H]I(1,4,5)P$_3$ (Fig. 2A); we identified the reaction product as a PP-[$^3$H]IP$_2$, because it was dephosphorylated back to [$^3$H]IP$_3$ by Ddp1 (the yeast ortholog of DIPP1), through specific cleavage of the β-phosphate (Fig. 2B).

### Structural analysis of four PP-IP isomers synthesized by *Tv*IPK

To determine structures of *Tv*IPK products, these were purified and soaked into crystals of human DIPP1; this experiment does not exclude the possibility that a substrate yields multiple products, only one of which is captured in DIPP1 crystal complexes. Four products were characterized using this approach: For example, we found that *Tv*IPK phosphorylates the 5-position of both *myo*-IP$_6$ and *myo*-(2OH)IP$_5$ to yield *myo*-5PP-IP$_5$ (i.e., 5-IP$_7$) and *myo*-5PP-I(1,3,4,6)P$_4$, respectively (Figure EV1A,B). This positional specificity imitates that of human IP6Ks (Draskovic et al, 2008; Wang et al, 2014). We similarly determined that *scyllo*-IP$_5$ is phosphorylated to *scyllo*-3-PP-I(1,2,4,5)P$_4$ (Figure EV1C), indicating that the latter's 3-6 axis substitutes for the 5-2 axis of *myo*-5-PP-Ins(1,3,4,6)P$_4$, such that both substrates have similar binding modes in which the orientation of the lone OH group (i.e., axial vs equatorial) has marginal impact on activity (Figs. 1D and EV1B,C, Appendix Fig. S2). We also determined that *scyllo*-L-I(1,2,3,4)P$_4$ is phosphorylated to *scyllo*-L-1,4-[PP]$_2$-I(2,3)P$_2$ (Figure EV1D). The second of these two pyro-phosphorylation events presumably

 

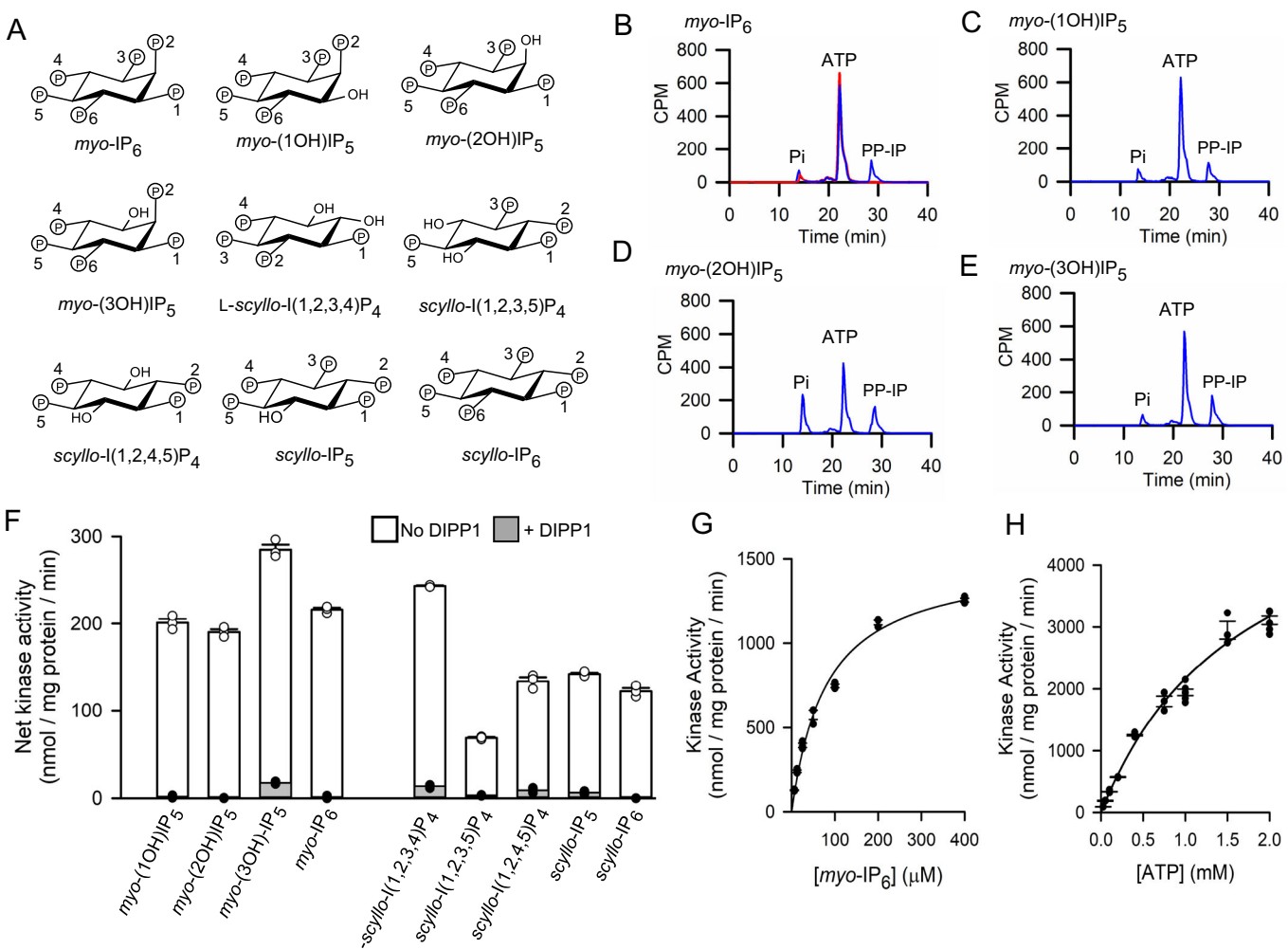

**Figure 1. Kinase activities of *TvIPK* in vitro.**

(A) Chemical structures of inositol phosphates incubated with *TvIPK* in the experiments described in this figure; phosphate numbering follows published guidelines (Murthy, 2006; Pramanik et al, 2020). See Appendix Fig. S1 for structures of other inositol phosphates used in this study. (B–E) Representative (1 of 3 independent experiments) HPLC data following incubations containing [γ-³³P]-ATP, either with *TvIPK* (blue traces) or without enzyme (red trace), plus 50 μM of (B) *myo*-IP₆ or (C) *myo*-(1OH)IP₅ or (D) *myo*-(2OH)IP₅ or (E) *myo*-(3OH)IP₅. (F) PP-IP product accumulation from assays performed either in the absence (open bars) or presence of *HsDIPP1* (closed bars) upon (i.e., apparent kinase activity) after incubation of *TvIPK* with 1 mM ATP and 200 μM of the indicated *myo*- or *scyllo*-IPs; (G) Substrate saturation plot for *TvIPK* incubated with 1 mM ATP and the indicated concentrations of *myo*-[³H]IP₆; activities were analyzed by HPLC. (H) Substrate saturation plot for *TvIPK* incubated with 2 mM *myo*-IP₆ and the indicated concentrations of [γ-³³P]-ATP; activities were analyzed by HPLC. (F–H) show each data point obtained from either 3 or 6 independent experiments (some data points are superimposed); standard errors are also indicated by vertical bars. Source data are available online for this figure.

requires the initial reaction product to either rotate within the catalytic pocket, or be released and rebound. In any case, these data provide further validation that *TvIPK* selectively adds a β-phosphate to pre-existing phosphate, and thus exclusively synthesizes PP-IPs.

## *TvIPK* activity inside a eukaryotic host

We next examined if *TvIPK* could synthesize PP-IPs in vivo, using *Saccharomyces cerevisiae* as a model system (Fig. 2C). Initially, we focused on the pathway of phosphorylation of *myo*-I(1,4,5)P₃. The latter is hard to detect in *myo*-[³H]inositol labeled wild-type cells but through its phosphorylation by the sequential actions of Arg82 and Ipk1, a peak of [³H]IP₆ is observed (Fig. 2D,E; and see Saiardi

et al, 2000b; York et al, 1999). The arg82Δ strain cannot phosphorylate *myo*-I(1,4,5)P₃, which therefore accumulates; [³H] IP₆ is not synthesized (Fig. 2D,E). Heterologous expression of *TvIPK* in the arg82Δ strain leads to the accumulation of more polar [³H]-labeled products migrating in the region of IP₄ and IP₅ standards, but fails to rescue IP₆ synthesis (Fig. 2A,B,D). This elution pattern resembles the profile reported for arg82Δ that overexpresses Kcs1, the yeast ortholog of IP6K (Dubois et al, 2002). Therefore, we posit that the more polar peaks generated by the action of *TvIPK* on IP₃ might represent PP-IP species such as PP-IP₂ and [PP]₂-IP; by possessing either four or five phosphate groups, respectively, these putative PP-IPs would be expected to elute in proximity to the IP₄ and IP₅ standards. To test this hypothesis, we prepared an extract of arg82Δ cells that express

**Table 1.** The relationship between substrate-binding mode and substrate-stimulated ATPase activity of *Tv*IPK.

| Substrate | Binding mode | Kinase activity (nmol/mg protein/min) | ATPase activity (nmol/mg protein/min) | ATPase activity (vs kinase activity) |
|---|---|---|---|---|
| *myo*-I(1,4,5)P$_3$ | BM2 | 12.6 ± 0.5 | Not detected | 0 |
| *myo*-(3OH)IP$_5$ | BM2 | 31.6 ± 0.8 | 5.7 ± 0.4 | 0.06 |
| *scyllo*-I(1,2,4,5)P$_4$ | BM2 | 34.8 ± 0.5 | 3.7 ± 0.4 | 0.11 |
| L-*scyllo*- I(1,2,3,4)P$_4$ | BM2 | 29.0 ± 0.3 | 1.17 ± 0.4 | 0.13 |
| *myo*-IP$_6$ | BM2 | 21.0 ± 0.7 | 2.9 ± 0.2 | 0.14 |
| *myo*-(1OH)IP$_5$ | BM1 | 18.0 ± 0.6 | 3.3 ± 0.2 | 0.18 |
| *scyllo*-IP$_6$ | BM1 | 24.2 ± 0.8 | 8.6 ± 0.2 | 0.36 |
| *scyllo*-IP$_5$ | BM1 | 32.8 ± 0.5 | 13.4 ± 0.3 | 0.41 |
| *myo*-(2OH)IP$_5$ | ND | 30.8 ± 0.7 | 22.6 ± 0.4 | 0.73 |
| D-*scyllo*- I(1,2,3,4)P$_4$ | BM1 | 4.0 ± 0.05 | 28.5 ± 0.3 | 7.1 |

Assays were performed with 50 μM [γ-$^{33}$P]-ATP and 50 μM of each substrate and were analyzed by HPLC. Within each HPLC run, the accumulation of [$^{33}$P]-Pi and [$^{33}$P]-labeled product (see Fig. 1B–E) was used to calculate phosphatase and kinase activities, respectively (data are means ± standard errors from 3 independent experiments). These kinase data are also presented graphically in the Appendix (Fig. S1). For the determinations of substrate binding modes, see Fig. 3G and Appendix Fig. S5; "ND" indicates that this particular binding mode was unable to be determined.

*Tv*IPK, and we incubated it with Ddp1 (Fig. 2E). This procedure decreased the levels of the putative PP-IP$_2$ and [PP]$_2$-IP peaks, while IP$_3$ accumulated (Fig. 2E). These data are consistent with *Tv*IPK synthesizing PP-IPs from IP$_3$ in vivo. Interestingly, *Tv*IPK also partly rescued the growth-inhibited phenotype of arg82Δ cells (Fig. 2C). Such data indicate that the absence of the canonical PP-IPs IP$_7$ and IP$_8$ can in part be functionally compensated by less phosphorylated PP-IP species, perhaps due to their relatively higher abundance. Finally, we found that loss of IP$_8$ in a vip1Δ strain was rescued upon expression of *Tv*IPK, by its phosphorylation of IP$_6$ through IP$_7$ to IP$_8$ (Fig. 2F).

## The overall structure of *Tv*IPK

To gain an atomic level understanding of the ability of *Tv*IPK to synthesize PP-IPs, we derived crystals of *Tv*IPK$^{17-265}$ in complex with ADP (1.95 Å resolution; Table EV1). The crystal structures were initially solved through a molecular replacement approach using human IPMK (PDB: 5W2G) as a template. For each asymmetric unit, there is one molecule of *Tv*IPK$^{17-265}$ in the C222$_1$ space group. The N-terminal residues 17–47 could not be traced, either in this crystal complex, or in a binary complex of full-length *Tv*IPK plus ADP, or in a number of *Tv*IPK$^{17-265}$/ADP complexes into which we soaked a variety of inositol phosphates (Table EV1, Figs. 3A,B and EV2A).

We have depicted a representative *Tv*IPK$^{17-265}$/ADP structural complex that also contains one of the most actively metabolized substrates, i.e., *scyllo*-IP$_6$, in a 6-equatorial chair conformation (Figs. 1F and 3B–G; Appendix Fig. S2). This substrate lies within a heart-shaped pocket (15 Å deep and 17 Å across at its widest point) that is lined with an array of side chains that are both positively charged and flexible (Fig. 3C–F). As far as we are aware (and see (Pramanik et al, 2020)), the conformation of *scyllo*-IP$_6$ has not previously been captured in any biological crystal complex. Note that all phosphates of *scyllo*-IP$_6$ are equivalent; to facilitate further discussion (see below) we have assigned position 3 to the phosphate group that is closest to the ADP β-phosphate. Note that a Pi molecule is also captured in this crystal complex (Fig. 3B,C). It is

possible this Pi is derived from the crystallization buffer and is retained through our soaking protocols. We also derived crystal complexes with ATP and AMP-PNP (see below), but only in the absence of IP substrates.

As stated above, *Tv*IPK is functionally related to IP6Ks. The overall *Tv*IPK fold is very similar to that of *Entamoeba histolytica* IP6KA (Figure EV2C), the only solved IP6K structure (Wang et al, 2014). However, there are some small but notable differences between the two structures. First, the viral N-lobe lacks three short β-strands found in *Eh*IP6KA (β6, β7, and β10). Second, the substrate binding loop in *Tv*IPK is much shorter and lacks the α-helical content found in the equivalent structural element of *Eh*IP6K, including the latter's 3$_{10}$ A helix, which is an important architectural element (Wang et al, 2014; Figs. 3A,B and EV2A; see below for further discussion). Third, *Tv*IPK exhibits a traceable N-terminal loop that includes 8 residues that are functionally equivalent to those expected for a so-called G-loop (Figure EV2A and see below); this element is missing from the *Eh*IP6KA structure (Wang et al, 2014).

## Characterization of nucleotide binding by *Tv*IPK

Kinases generally utilize Mg$^{2+}$ to facilitate nucleotide and substrate binding, catalysis, and product release (Knape et al, 2017), but we were unable to derive Mg$^{2+}$/ATP/*Tv*IPK crystal complexes. However, nucleotide/kinase crystal complexes can sometimes be derived by replacing Mg$^{2+}$ with another divalent metal ion such as Cd$^{2+}$ (Knape et al, 2017). Indeed, by making this substitution, we derived an ATP/*Tv*IPK crystal complex.

Deconvolution of the nucleotide's electron density revealed two alternate conformations, which we have named ATP-A (55% occupancy) and ATP-B (45% occupancy) (Fig. 4A). The ATP-B triphosphate has an overextended (Buelens et al, 2021) configuration (i.e., the α–β–γ phosphate angle is 134°), enabling its unique γ-phosphate interactions with Arg58 plus the phosphate molecule (Figure EV3A,B). This ATP-B configuration is replicated by AMP-PNP, perhaps facilitated by the latter's more rigid NH group (Figure EV3C–E). In contrast, for ATP-A, the triphosphate group is

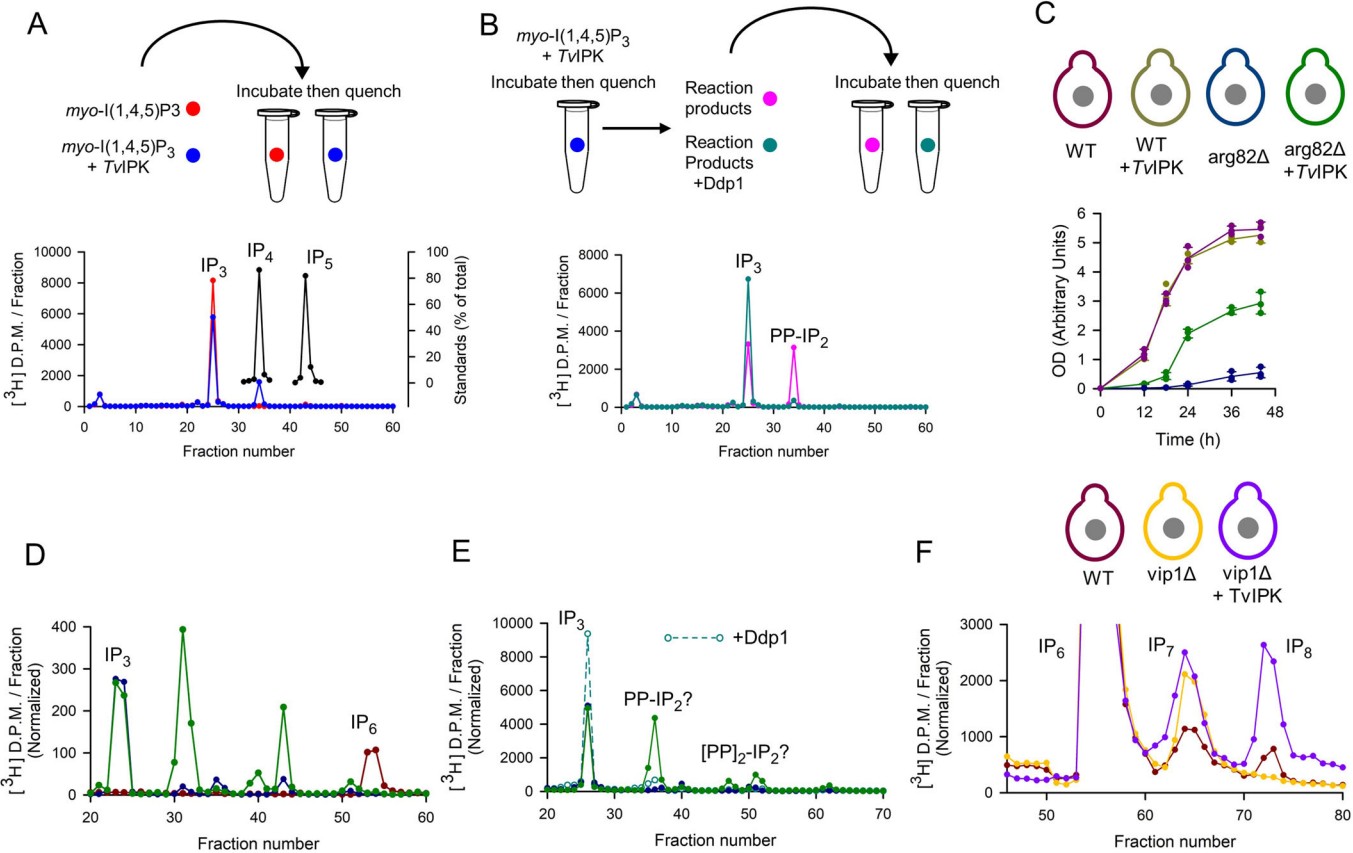

**Figure 2. Phosphorylation of *myo*-IPs by *Tv*IPK, both in vitro and when expressed in *S. cerevisiae*.**

(A) The upper panel is a graphical representation of an experiment in which *myo*-[³H]I(1,4,5)P₃ was incubated in vitro either with (blue symbols) or without (red symbols) recombinant *Tv*IPK. The lower panel shows the corresponding HPLC analyses (blue and red plots) plus separately determined elution of [³H]-labeled standards of *myo*-I(1,3,4,5)P₄ and *myo*-(2OH)IP₅ (black plots). (B) The upper panel is a graphical representation of an experiment in which [³H]-labeled reaction product (from the experiment described in panel A), which coeluted with the [³H]IP₄ standard, was purified and incubated with (green plot) or without (purple plot) Ddp1. The lower panel shows HPLC analyses of these incubations. See text for reasoning behind labeling the two peaks as IP₃ and PP-IP₂. (C) Graphic indicating the color coding of four strains of *S. cerevisiae*: wild-type (dark purple), wild-type + *Tv*IPK (dark yellow), arg82Δ (dark blue) and arg82Δ + *Tv*IPK (green). Growth plots are also depicted for each of these yeast strains (each data point from 3 separate experiments are shown; standard errors are also indicated by vertical bars). (D) HPLC analysis of extracts prepared from three yeast strains (see (C) for color codes) that had been labeled with *myo*-[³H]inositol. (E) HPLC analysis of extracts prepared from *myo*-[³H]inositol-labeled arg82Δ cells (dark blue plot) and arg82Δ cells hosting *Tv*IPK (green symbols) and also an extract from arg82Δ cells hosting *Tv*IPK that had been incubated with Ddp1 (green trace; open symbols/broken lines). (F) Graphic indicating the color coding of three strains of *S. cerevisiae*: wild-type (dark purple), vip1Δ (yellow) and vip1Δ + *Tv*IPK (purple). Each of these strains were labeled with [³H]-inositol and then cell extracts were prepared for HPLC analysis (the IP₆/IP₇/IP₈ region of the chromatograph). All HPLC data are representative of 3 independent experiments. Source data are available online for this figure.

relatively compact (i.e., the α–β–γ phosphate angle is 115°). In both ATP configurations, Cd atoms form an intense network of salt bridges with all three phosphates, the bridging oxygens, Lys72 from the N-lobe, and two C-lobe residues, Ser233 and Asp231 (Figs. 4B,C and EV3A,B). The latter additionally makes polar contact with Lys72 (Fig. 4B,C), and so this primitive enzyme contains all components of the highly conserved IPK "catalytic triad" (Lys/catalytic Asp/metal (Shears and Wang, 2019).

We noted that Pi (perhaps arising from the crystallization buffer, see above) is present within the crystal complexes that contain either ATP or AMP-PNP (Figs. 4A,B and EV3A–E); the relative positioning of the Pi is different in each case (Figure EV3F). Both positions are themselves different from that of the Pi within the *Tv*IPK/*scyllo*-IP₆ crystal complex (Figs. 3B,C and EV3F). These data are suggestive of conformational flexibility of the Pi-binding site.

Figure 4A describes the near-exact superimposition of the adenine and ribose moieties from both ATP configurations A and B. These groups are docked in a hydrophobic pocket that is surrounded by residues Arg54 from the G-loop, Phe52, Ile61, Ile70, Pro100 and Leu112 from the N-lobe, Asp114 and Thr116 from the hinge region, and Val124, Ile215 and Val230 from the C-lobe (Fig. 4C). The $N^1$ and $N^6$ atoms of adenine form two hydrogen bonds with the amide nitrogen of Val115 and the carbonyl oxygen of Glu113, respectively (Fig. 4B,C). The ribose oxygens make hydrogen bonds with the G-loop (the amide nitrogen of Ile55 and Ala56) as well as Asp126 from the C-lobe. The adenine ring itself is sandwiched between three layers of hydrophobic residues in the N-lobe and three layers from the C-lobe, thereby forming a stabilizing, architectural "C-spine" (Fig. 4D). A second, smaller "R-spine" comprises just 4 hydrophobic residues, anchored to a loop proximal to the α5 helix through a hydrogen bond between Asp242 and the backbone amide of Phe209 (Fig. 4E). Both of

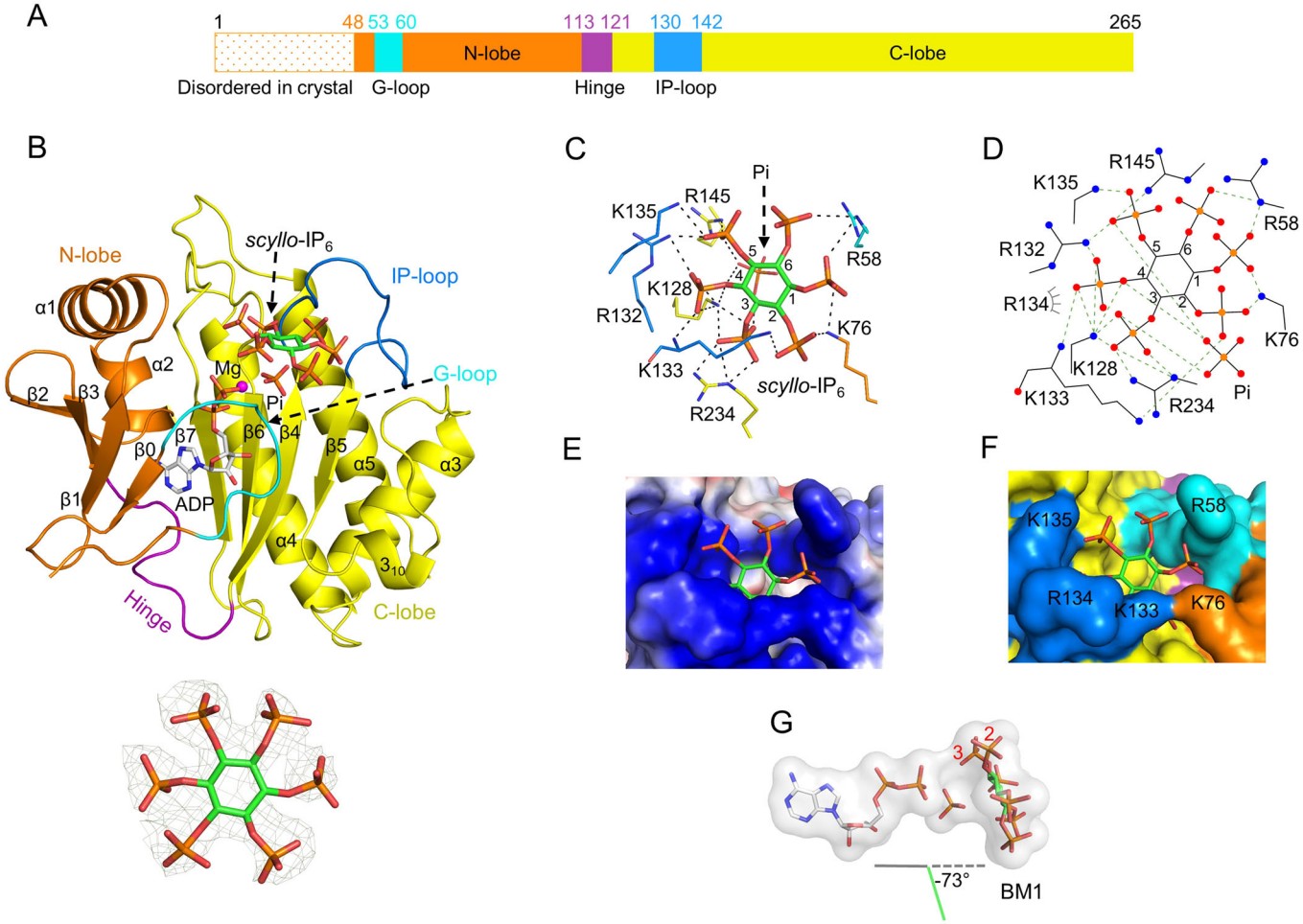

**Figure 3. The *Tv*IPK/ADP/*scyllo*-IP$_6$ crystal complex.**

(A) Domain graphic of full-length *Tv*IPK. The G-loop is colored cyan, the N-lobe is colored orange, the hinge region is colored purple, the IP-loop is colored blue and the C-lobe is colored yellow; this color scheme is retained in subsequent figures. (B) Ribbon plot to depict the structure of the *Tv*IPK/ADP/*scyllo*-IP$_6$ crystal complex. The ADP and *scyllo*-IP$_6$ are depicted in stick format: gray for carbon in ADP, green for carbon in *scyllo*-IP$_6$, blue for nitrogen, orange for phosphorus, red for oxygen. Magnesium atoms are depicted as magenta spheres. The substrate's Fo-Fc electron density map (green mesh) is contoured at 2.5 σ. (C, D) polar contacts (broken lines) within the catalytic pocket, depicted in stick format or rendered with Ligplot+, respectively. Van der Waals contacts are shown as "eyelash" graphics in D panel. Note that all *scyllo*-IP$_6$ phosphates are equivalent; to facilitate discussion, the phosphate group that is closest to the nucleotide is assigned position 3 because that is the group in *scyllo*-IP$_5$ that is pyrophosphorylated (see text and Fig. 5 for further details). (E, F) the ligand binding pocket is depicted as electrostatic surface or space-filling representations, respectively. (G) Space-filling model of the relative positions of ADP, Pi and *scyllo*-IP$_6$. Note the graphical representation of the angle (−73°) at which the plane of the inositol ring intersects a second plane that runs between the α- and β-phosphorus atoms of ADP through the bridging oxygen. BM1 binding mode 1. For further details see the main text and Figure EV3.

these architectural elements were initially discovered from studies of protein kinases, and were recently found to be present in IPKs (Shears and Wang, 2019); the conservation of these spines in the viral IPK speaks strongly to their overall significance. It is of further interest that the positioning of the side-chain of Leu112 between the two spines establishes the depth of the nucleotide-binding pocket, and hence fulfils the role of a "gatekeeper"; such residues may also have a regulatory function (Shears and Wang, 2019).

*Tv*IPK contains an N-terminal G-loop ([53]PRIAGRSY[60]) that has extensive polar contacts with both the adenosine and triphosphate moieties of ATP (Fig. 4B,C,F–H). A spatially and functionally analogous loop has been characterized in in mouse (Mus musculus) *Mm*IP3KB (Chamberlain et al, 2005) and IP5 2K (Franco-Echevarria et al, 2017; Gonzalez et al, 2010) and also in protein

kinases, in which it was first established that the role of this loop is to establish nucleotide specificity and to control reaction rates (Barouch-Bentov et al, 2009). Indeed, Arg58 in the *Tv*IPK G-loop interacts with the ATP phosphates (Figs. 4B,C and EV3A,B). Interestingly, the G-loop in IP5 2K crystal complexes (PDB: 5MW8 and 2XAN) is shorter (4 residues) but closer to the nucleotide (e.g., Figure EV3G), also facilitating extensive ionic interactions with ATP phosphates (Franco-Echevarria et al, 2017; Gonzalez et al, 2010). In contrast, the residue in *Mm*IP3KB that corresponds to Arg58 in the *Tv*IPK is His672 (Fig. 4F), which occupies the β-phosphate space in the apo-enzyme, but this His residue does not contact ATP in the *Mm*IP3KB/nucleotide crystal complex (Chamberlain et al, 2005), in part because the IP3KB G-loop in 1.9 Å displaced relative to the corresponding loop in *Tv*IPK (Fig. 4G).

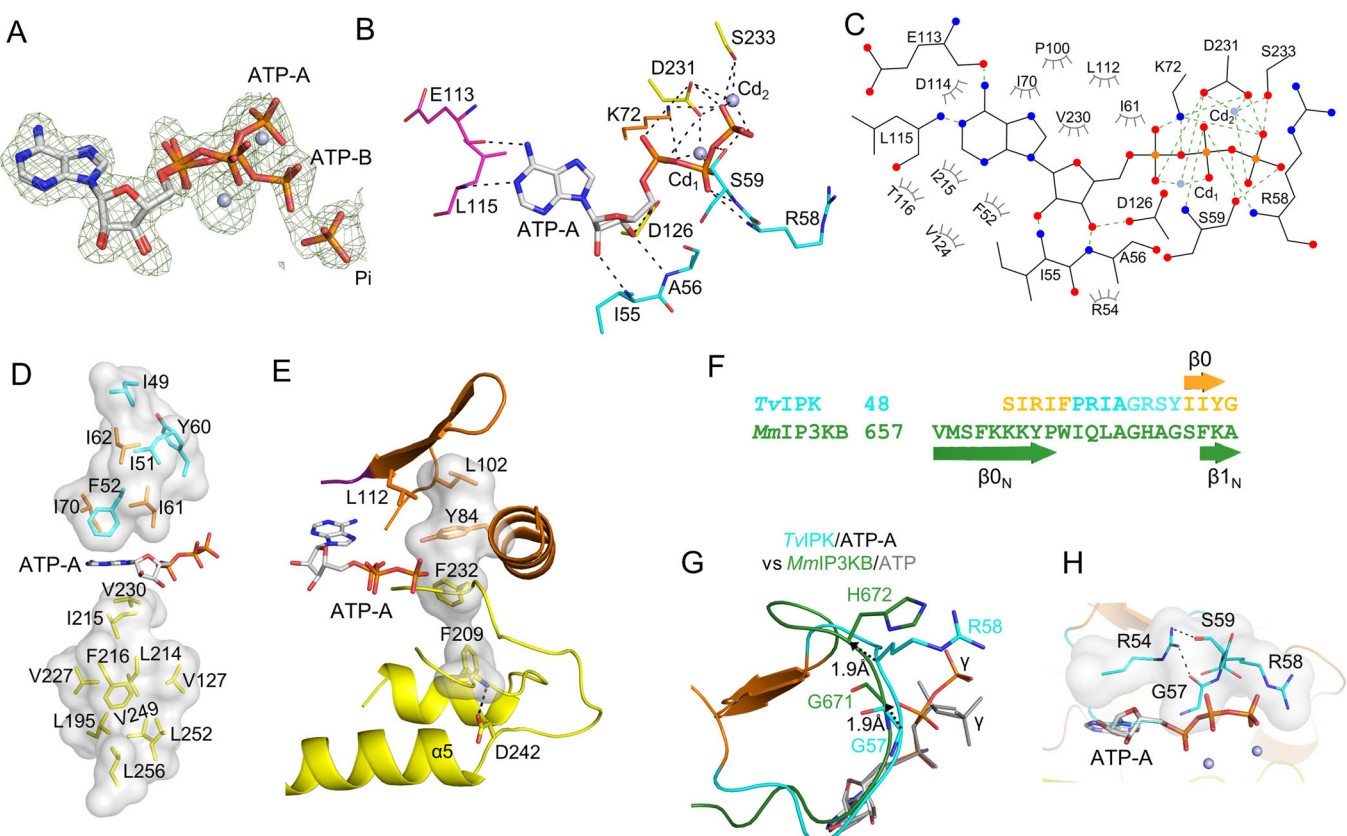

**Figure 4. The *Tv*IPK/ATP/Cd crystal complex.**

(A) The Fo-Fc electron density map for ATP, Cd and Pi (green mesh), contoured at 3 σ. The two conformations of ATP that are mapped by the electron density are shown as stick models and labeled ATP-A and ATP-B. Cadmium atoms are depicted as light blue spheres. Pi is shown in stick format. (B) Stick model of *Tv*IPK residues (color coded as in Fig. 3) that participate in polar contacts (broken lines). (C) Protein/protein and protein/ligand interactions rendered with Ligplot+; Van der Waals contacts are shown as "eyelash" graphics. (D, E) Ribbon plot to depict the hydrophobic C-spine and R-spine in *Tv*IPK, respectively; key residues are shown as space-filling representations and stick-models, in which bound nucleotides were superimposed, colored as in Fig. 3B. ATP-A is shown as a stick model. (F) structure-based sequence alignment of the G-loops of *Tv*IPK and *Mm*ITPKB. (G) Stick model superimposition of the G-loops from *Tv*IPK (blue, in complex with ATP-A) and *Mm*IP3KB (green; in complex with ATP; PDB: 2AQX), represented as tubes that trace Cα atoms, except for the stick models of the indicated side chains. (H) Space-filling rendition of the *Tv*IPK G-loop covering the nucleotide binding pocket; intra-protein polar interactions are depicted as broken lines.

Gly57 in *Tv*IPK permits the close approach of the ATP γ-phosphate to coordinate with Arg58 (Fig. 4H). An intramolecular hydrogen bond between Arg54 and the Gly57 backbone may contribute to loop stabilization (Fig. 4H). We validated the catalytic importance of Arg58 and Gly57 by mutating each of them to Ala, which inhibited kinase activity by >95% with *myo*-IP$_6$ as the substrate (Table 2).

Additionally, Ile55 and Ala56 in *Tv*IPK form hydrogen bonds with the ribose moiety of ATP; the functionally equivalent G-loop residues in *Mm*IP3KB are Leu669 and Ala670 (Chamberlain et al, 2005). In contrast, the G-loops in IP5 2Ks do not interact with the ribose group (Franco-Echevarria et al, 2017; Gonzalez et al, 2010).

## Substrate-binding by *Tv*IPK

Including *scyllo*-IP$_6$ (see above), we present nine enzyme/ADP crystal complexes that contain *myo*- or *scyllo*-IPs (Figs. 3B–G and EV4A–H). In all but two of these complexes, each IP substrate makes extensive polar contacts with the same nine positively charged residues that are distributed through several regions of the

protein: Arg58, Lys76, Lys128, Arg132, Lys133, Arg134, Lys135, Arg145 and Arg234 (Figs. 3C,D and EV4A–H). The flexibility of these side chains seems likely to contribute to the overall pliability required for binding of multiple substrates, aided by the high degree of structural symmetry between many of these IPs (Marquez-Monino et al, 2021; Zong et al, 2021). The relative functional significance of each of these residues was investigated by single-site mutagenesis, using *myo*-IP$_6$ as the substrate. Except for the R134A mutation (14% inhibition), every other mutant protein showed significantly compromised kinase activity (74% to 99% inhibition; Table 2).

From our substrate/enzyme crystal complexes, two substrate binding modes (BM) were identified (i.e., BM1 and BM2). These were conveniently distinguished by determining the angle at which the plane of the inositol ring intersects a second plane that runs between the α- and β-phosphorus atoms of ADP through the bridging oxygen: this intersection approached the perpendicular for BM1 (consistently −73° to −70°) whereas it is close to horizontal for BM2 (i.e., a more variable range of +10° to +33°) (Figs. 3G and EV4A–H). The nature of the captured binding mode did not

**Table 2. Single site mutagenesis of *Tv*IPK: impact on kinase activity against *myo*-IP₆.**

|  | Activity (nmol/mg protein/min) |
|---|---|
| WT | 246.6 ± 5 |
| R54A | 50.6 ± 5.4 |
| G57A | 4.6 ± 0.4 |
| R58A | 11.6 ± 0.6 |
| K76A | 22.4 ± 0.9 |
| K128A | 7.3 ± 0.7 |
| R132A | 0.6 ± 0.1 |
| K133A | 32.6 ± 0.7 |
| R134A | 213.5 ± 1 |
| K135A | 65.7 ± 1.5 |
| R145A | 58.4 ± 1 |
| R234A | 2.7 ± 0.2 |

Assays were performed with 50 μM [γ-$^{33}$P]-ATP and 50 μM of each substrate and were analyzed by HPLC. Data represent means and standard errors from 3 independent experiments. Student's *t* test was used to determine statistical significance of each mutant vs wild type enzyme. For R134A, $p = 0.03$. For the other mutants, $p < 0.002$.

exhibit a relationship with the degree of kinase activity (Table 1). On the other hand, BM1 is associated with higher ratios of ATPase activity versus kinase activity, as compared to BM2 (Table 1).

We next interrogated our data for evidence that any of our enzyme-bound substrates might be in catalytically productive orientations to support the kinase positional specificity that we have identified. Among the four reaction products that we have characterized (Figure EV1A–D), only three of the corresponding precursor substrates were captured in crystal complexes: *myo*-IP₆, *scyllo*-IP₅ and L-*scyllo*-I(1,2,3,4)P₄ (Figure EV4D,F,H). A model of the orientation of *myo*-IP₆ that is superimposed upon either ATP-A or ATP-B cannot account for the phosphorylation at the 5-position (Figure EV5A,B) that is required to yield the *myo*-5PP-IP₅ product (Figure EV1A). Equally, a model of the orientation of L-*scyllo*-I(1,2,3,4)P₄ that is superimposed upon either ATP-A or ATP-B cannot account for its phosphorylation at the 1- and 4-positions (Figure EV5C,D) that would generate the L-*scyllo*-1,4[PP]₂-I(2,3)P₂ product (Figure EV1D). Superimposition of either ATP-A or ATP-B upon *scyllo*-IP₅ (Fig. 5A,B) indicated that in neither case is the 3-phosphate suitably aligned for in-line phosphorylation that yields the *scyllo*-3-PP-I(1,2,4,5)P₄ product (Figure EV1C).

On the other hand, in the case of ATP-A, a rotation of the inositol ring by 60° would bring the γ-phosphate and 3-phosphate of *scyllo*-

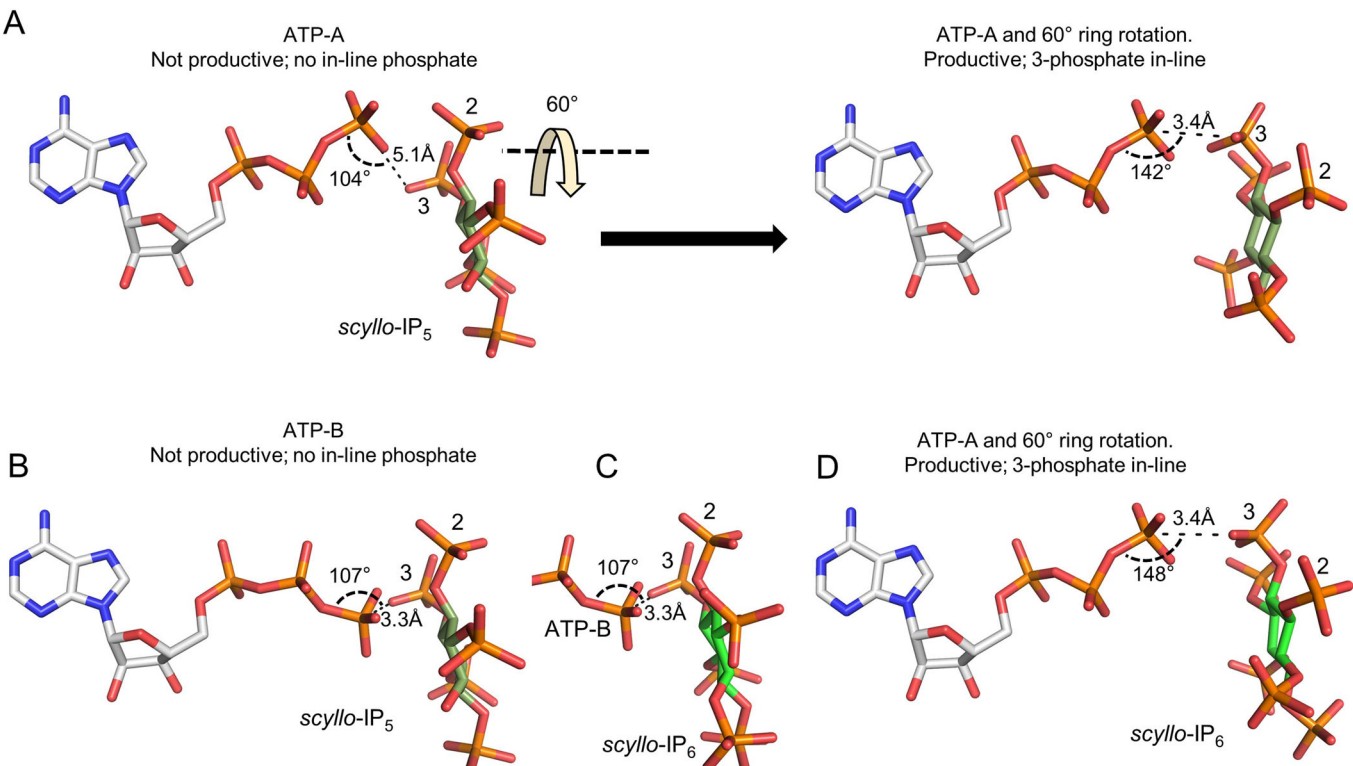

**Figure 5. A proposed productive binding mode for *scyllo*-IP₅ and *scyllo*-IP₆.**

(A) Left-hand panel: superimposition of *scyllo*-IP₅ with ATP in conformation A. In the right-hand panel the inositol ring is rotated 60° without modification to the plane of the ring. Also indicated is the angle formed by the attacking oxygen, γ-phosphorus and the bridging oxygen (142°) and the nucleophilic attack distance of 3.4 Å (broken line). (B) Superimposition of ATP in conformation B and *scyllo*-IP₅ in stick model. (C) The orientation of *scyllo*-IP₆, as determined by the parameters used to generate the orientation of *scyllo*-IP₅. (D) The inositol ring shown in (C) is rotated 60° without modification to the plane of the ring. Also indicated is the angle formed by the attacking oxygen, γ-phosphorus and the bridging oxygen (148°) and the nucleophilic attack distance of 3.4 Å (broken line). Note that all *scyllo*-IP₆ phosphates are equivalent; to facilitate discussion, the phosphate group that is closest to the ATP-A γ-phosphate is assigned position 3, because that is the group in *scyllo*-IP₅ that is pyrophosphorylated.

IP$_5$ into an alignment (3.4 Å apart with a 142° angle) that is potentially supportive of an in-line nucleophilic reaction (Fig. 5A). This analysis leads us to posit that we have captured scyllo-IP$_5$ substrate in a pre-reaction configuration, that potentially could be reoriented into a catalytically permissive orientation, perhaps assisted by the flexibility of the positively charged side chains of the amino-acid residues in TvIPK that line the catalytic pocket.

It is apparent that there are very similar orientations of scyllo-IP$_5$ and scyllo-IP$_6$ within the catalytic pocket, indicating the latter also cannot be phosphorylated by ATP-B (Fig. 5B,C). Again, a rotation of the inositol ring by 60° would bring the ATP-A γ-phosphate and the scyllo-IP$_6$ acceptor phosphate into an alignment (3.4 Å apart with a 148° angle) that is potentially supportive of an in-line nucleophilic reaction (Fig. 5D).

# Discussion

Our discovery and characterization of a minimalist IPK encoded by a NCLDV genome represents a significant advance for several reasons. First, we demonstrate this enzyme synthesizes PP-IPs, which represent the most evolutionarily ancient members of the IP family of intracellular signals (Bennett et al, 2006; Saiardi, 2012). Indeed, there is evidence that the emergence of NCLDVs predates the origin of eukaryotes (Guglielmini et al, 2019). Nevertheless, the distinct evolutionary trajectories of these viruses and their hosts has licensed NCLDVs as independent evolutionary innovators that subsequently returned modified genomic content to pro-eukaryotes and early eukaryotes, possibly contributing to the early success of this lineage (Guglielmini et al, 2019; Moniruzzaman et al, 2020). This pro-active and two-way evolutionary exchange of genomic material counters the traditional view that viruses merely pickpocketed (Moniruzzaman et al, 2020) and then de-evolved cellular DNA solely for their own exploitation. Thus, our study offers alternate directions for understanding the very early evolution of PP-IPs, the kinases that synthesize them, and their primordial roles in signal transduction. These advances testify to the value of taking an evolutionary approach to furthering understanding of IP/PP-IP biology.

Our structural characterization of a small, viral IPK is informative for uncovering the considerable complexity of a relatively small protein sequence. In TvIPK we have observed many of the catalytic refinements of the IPK family—and the related protein kinase family—including the D[I/V]K[I/L]G sequence that stabilizes a transition state, and the important [V/I]DF tripeptide (Appendix Fig. S1). In the latter case, the Asp coordinates a catalytically essential Mg, and the Phe is a key component of a one of two hydrophobic architectural spines that are stabilizing entities in all IPKs (Shears and Wang, 2019). Analysis of TvIPK has also improved insight into the G-loop, which positions the ATP phosphate groups, so as to establish nucleotide specificity and control reaction rates.

We suggest that minor conformational changes are also required to rotate the plane of the inositol ring by 60° to establish catalytic productivity for two substrate configurations that we have captured in our TvIPK crystal complexes: those for scyllo-IP$_5$ and scyllo-IP$_6$ (Fig. 5). This proposal is consistent with our demonstration that TvIPK pyrophosphorylates the 3-phosphate of scyllo-IP$_5$ (Figure EV1). It is interesting to contrast the relatively robust activity of TvIPK against scyllo-IP$_5$ and scyllo-IP$_6$ with the negligible activity

that human IP6K1 exhibits against both scyllo-IP$_5$ and scyllo-IP$_6$ (Mohanrao et al, 2021), and the inability of mammalian IP5 2 K to phosphorylate scyllo-IP$_5$ (Riley et al, 2006). Perhaps mammalian IP6Ks have evolved to maximize phosphorylation of myo-IPs and not scyllo-IPs, given there is no evidence the latter are present in mammalian cells. In contrast, both scyllo- and myo-IPs are abundant in soils (Reusser et al, 2020), and amoeba have the potential to accumulate them by phagocytosis, whereupon they could naturally be available to TvIPK after viral infection. Of course, the eukaryotic-wide distribution of myo-IPs signaling cascades also provides an endogenous supply of substrates in any Terrestrivirus-infected amoeboid host. Consequently, we can expect that both myo- and scyllo-PP-IPs will be released into the soil upon cell lysis. Exploration of the significance of PP-IPs in the soil niche could be a productive new area of future research, not in the least because the increased negative charge brought about by pyrophosphorylation could facilitate their adsorption onto solid matrices or their precipitation, thereby locking them out of geochemical Pi cycles (Celi and Barberis, 2007).

Besides scyllo-IP$_5$ and scyllo-IP$_6$, we captured 7 additional substrate configurations (Figure EV4), but we do not have sufficient information to determine if these might also be pre-reaction complexes, or merely non-productive binding modes. This is a recurring complexity in the field, due to the high degree of structural symmetry between many IPs (Marquez-Monino et al, 2021). Nevertheless, we were able to designate just two binding modes (BM1 and BM2), based upon the angle at which the plane of the inositol ring intersects a second plane that runs between the α- and β-phosphorus atoms of ADP through the bridging oxygen (Figs. 3G and EV4A–H). Interestingly, EhIP6KA also retains two substrate binding modes, one of which is not productively oriented towards the ATP γ-phosphate (Wang et al, 2014). These observations with TvIPK and EhIP6KA observations add emphasis to the need (mentioned above) to unravel conformational dynamics of IPKs in order to fully rationalize their catalytic activities.

How could the synthesis of PP-IPs by TvIPK support viral propagation? PP-IPs generally act at a cell-signaling interface with molecular processes that direct metabolic and bioenergetic homeostasis (Azevedo and Saiardi, 2017; Shears, 2018), particularly with regards to sustaining cellular levels of Pi and ATP (Gu et al, 2017; Riemer et al, 2021; Szijgyarto et al, 2011; Wild et al, 2016). Thus, PP-IPs can help facilitate the hypermetabolic state of the host cells that is critical for viral biogenesis (Brahim Belhaouari et al, 2022). Consistent with this concept is our demonstration that TvIPK supports the energetically demanding process of yeast cell proliferation (Fig. 2F). These ideas add substantially to our insights into viral interactions with IP-signaling cascades inside eukaryotic cells, which previously centered on the role of myo-IP$_6$ in promoting maturation of HIV capsids (Mallery et al, 2021).

The efficiency of PP-IP generation by TvIPK in vitro (Fig. 1) and in vivo (Fig. 2) begs the question as to how widespread this enzyme might be among NCLDVs. In addition to Terrestrivirus, we have identified two additional NCLDV genomes that encode candidate IPKs: Indivirus and Barrevirus (Appendix Fig. S1). However, it is likely viral IPK diversity is still underestimated; recent literature (Queiroz et al, 2022; Schulz et al, 2022) predicts that only a small fraction of both total NCLDV genomes and their amoeboid hosts have been identified. We should also take into account the likelihood of discovering hitherto unrecognized NCLDVs in thawing permafrost

(Alempic et al, 2023; Jansson and Wu, 2022). We believe our data will help stimulate expansion of these areas of research, and further illuminate the biogeochemistry of environmental phosphate cycles, which are not just of immediate environmental concern, but also necessary for long term food security due to imminent depletion of geological phosphate reserves.

# Methods

## Protein expression and purification

The codon-optimized cDNA of *Tv*IPK (GenBank: AYV76578.1, Appendix Table S1) was purchased from Genscript Inc and subcloned into the pDest-566 vector. This vector encodes a His6 tag, maltose-binding protein tag and tobacco etch virus protease cleavage site at the N terminus. Mutants were prepared using a site-directed mutagenesis kit (Stratagene) or Q5 site-directed mutagenesis kit (Biolabs) (Appendix Table S1); all mutants were verified by sequencing. The recombinant plasmid was transformed into competent *E. coli* BL21(DE3). An overnight culture of the transformed *E. coli* cells was inoculated into nutrient-rich 2×YT medium (16 g/L Tryptone, 10 g/L yeast extract and 5 g/L NaCl at pH 7.5) which was cultured at 37 °C to an optical density of 0.7 at 600 nm. Isopropyl β-D-thiogalactopyranoside (0.1 mM) was then added and cultures were continued at 15 °C for 22 h.

*E. coli* cells were harvested by centrifugation at $5000 \times g$ for 10 min and disrupted using a Constant Cell Disruption System (Constant System Ltd) at 20 KPsi. Recombinant protein was then purified at 4 °C as follows: cell supernatant was mixed with Ni-NTA agarose (Qiagen), then washed with buffer containing 300 mM NaCl, 20 mM Tris-HCl, pH 7.2, 20 mM imidazole, and the target protein was eluted with imidazole concentration of 250 mM. The eluate was loaded to HiTrap™ Heparin HP column (Cytiva) and eluted with 10 column volumes of a 50–2000 mM NaCl gradient in 20 mM Tris-HCl, pH 7.2. After removing the maltose-binding protein tag with tobacco etch virus protease, the protein was further purified with another HiTrap™ Heparin HP column. Finally, the protein was applied to a Superdex™ 200 size exclusion column (Cytiva) and eluted with 150 mM NaCl, 20 mM Tris-HCl pH 7.2. The final preparations were concentrated to 7 mg/mL and stored at −80 °C; purity (>95%) was validated by SDS-PAGE.

## Enzyme assays

Unless otherwise stated, the enzyme assays were performed at 25 °C for 30–90 min in 100 µL reaction mixtures containing 0.3–60 µg/mL wild-type *Tv*IPK, 20 mM HEPES pH 7.2, 100 mM KCl, 3.5 mM MgCl_2, 0.01% TritonX-100, 50 µM Na_2EDTA, 50 µM ATP (Gold Biotechnology, Catalog number A-081-1) plus 0.01 to 0.02 µCi [γ-$^{33}$P]ATP, at concentrations of either 50 µM or 200 µM as indicated, and 50 µM *myo*-IP substrates (purchased from either Sigma Aldrich or Cayman Chemical Company), or *scyllo*-IPs synthesized as previously described (Kwon et al, 2003). In some assays, 0.7 mg/mL *Hs*DIPP1 prepared as previously described (Zong et al, 2021), was added to the reaction system as described in the figure legends.

Reaction kinetics were determined for the *myo*-IP_6 and ATP, at various concentrations as described in the figure legends, and were analyzed using GraphPad Prism. 3U creatine kinase and 2.5 mM creatine phosphate were added into the reaction system. Mutant versions of *Tv*IPK (0.5–10 µg/mL) were assayed as described above in the presence of 1 mM ATP-Mg and 10 µM IP_6, plus $1 \times 10^5$ DPM $^3$H-IP_6.

Reactions were quenched by addition of 200 µL of 0.3 M NH_4H_2PO_4 (pH 4.0) plus 40 mM EDTA and analyzed by using a $4.6 \times 250$ mm Synchropak Q100 HPLC column. The elution gradients (1 mL/min) were generated by mixing Buffer A (1 mM Na_2EDTA) with Buffer B (Buffer A plus 2.5 M NH_4H_2PO_4, pH 4.0). For gradient 1, to separate InsP_4 products: 0–1 min Buffer A, 1–21 min 40% buffer B. For gradient 2: 0–1 min Buffer A, 1–21 min 70% buffer B. The eluate was mixed with 0.5 mL/min Monoflow4 scintillation liquid (National Diagnostics) and radioactivity was monitored in-line with a Beta-RAM 6 radio flow detector (LabLogic) using Laura6 data collection software.

## Crystallization and crystal soaking

Either the full length wild-type *Tv*IPK protein, or a truncated construct (*Tv*IPK$^{17-265}$) were added at a concentration of 7 mg/mL to 5 mM ADP and 10 mM MgCl_2 and incubated on ice for 30 min, then crystallized by hanging drop vapor diffusion against a well buffer of 12% PEG8000, 100 mM HEPES pH 7.0, 10 mM NaH_2PO_4 and 10% ethylene glycol at 4 °C. The crystals of *Tv*IPK$^{17-265}$ were then soaked in 25% PEG8000, 100 mM HEPES pH 7.0, 20% ethylene glycol with 5 mM IPs, 5 mM ADP and 10 mM MgCl_2 overnight at 4 °C. The complex structures of *Tv*IPK/ATP and *Tv*IPK/AMP-PNP were obtained by substituting MgCl_2 and ADP in the soaking buffer with 2 mM CdCl_2 and 5 mM of ATP (or AMP-PNP), respectively.

To identify reaction products of *Tv*IPK kinase activity, 1 mM of each IP substrate was incubated with 1 mg/mL *Tv*IPK at 37 °C for 12 h in 100 µl reaction buffer containing 20 mM HEPES pH 7.0, 150 mM KCl, 2 mM MgCl_2, 50 µM Na_2EDTA and 3 mM ATP. The reactions were dried under a vacuum in room temperature, and then 10 µL of soaking buffer (25% PEG8000, 100 mM HEPES pH 7.0, 20% ethylene glycol) was added. After centrifugation, the resulting supernatant was added to crystals of *Hs*DIPP1 and soaked for 12 h at 25 °C.

## Structural data collection, analysis, and refinement

Diffraction data were collected using APS beam lines 22-ID and 22-BM. All data were processed with the program HKL2000 (Otwinowski and Minor, 2007). The initial structure of *Tv*IPK was determined by molecular replacement with *Hs*IPMK as the template (PDB: 5W2G) using the phaser module in PHENIX (Adams et al, 2010). The initial structure was manually rebuilt with COOT (Emsley and Cowtan, 2004) and refined with PHENIX. The molecular graphics representations were prepared with the program PyMol (Schrödinger, LLC). Atomic coordinates and structure factors have been deposited with the Protein Data Bank (Table EV1).

## Generation of yeast strains expressing *Tv*IPK

The yeast codon-optimized nucleotide sequence for *Tv*IPK was chemically synthesized and cloned into the plasmid pCA82 (Azevedo et al, 2020) using SalI and NotI restriction sites; the

pCA82 was derived from the commercially available Invitrogen pYes3 plasmid in which the inducible galactose promoter (GAL1) has been substituted with the constitutive alcohol dehydrogenase promoter (Azevedo et al, 2020). The resulting vector and the empty vector were transformed into wild-type yeast (BY4741) and the corresponding arg82Δ knockout strain using the lithium-acetate protocol (Bosch and Saiardi, 2012). The growth assay was performed in Synthetic Complete (SC) media minus tryptophan (Formedium).

## Labeling of yeast strains with *myo*-[³H]inositol and IP profiling

Labeling of yeast cultures with *myo*-[³H]inositol and extraction of inositol phosphates were performed as described previously (Azevedo and Saiardi, 2006). Briefly, yeast precultures in synthetic complete media without tryptophan (SC-TRP) were used to inoculate 5 mL of SC-TRP-Inositol containing 5 µCi/mL of *myo*-[³H]-inositol. Cells were grown overnight to logarithmic phase at 30 °C, harvested by centrifugation (2 min, 2000 × g), and washed once in water. IPs were extracted by adding glass beads and 1 M perchloric acid containing 3 mM EDTA. Cells were broken by shaking with a vortex (5 min, 4 °C) and debris were removed by centrifugation (5 min, 15,000 × g). The supernatant was neutralized by adding potassium carbonate. Samples were kept on ice for 2 h to allow for salt precipitation. Insoluble material was removed by centrifugation (5 min, 15,000 × g) and the supernatant was analyzed by SAX-HPLC as reported (Azevedo and Saiardi, 2006). The neutralized yeast extract was supplemented with 1/10 volume of 10× assay buffer to yield final concentrations as follows: 20 mM HEPES pH 7.0, 100 mM KCl, 6 mM MgCl₂, 1 mM DTT. Next, we added approximately 20 ng of recombinant Ddp1 (purified as described by Lonetti et al, 2011) and incubated the samples at 30 °C for 1 h, then reactions were directly analyzed by SAX-HPLC.

## Data availability

All of the structural data have been deposited in the Protein Data Bank http://www.wwpdb.org with the following links: 8T8U, 8T8V, 8T8W, 8T8X, 8T8Y, 8T8Z, 8T90, 8T91, 8T92, 8T93, 8T95, 8T96, 8T97, 8T98, 8T99, 8TF9, 8TFA.

## Peer review information

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

## Acknowledgements

This research was supported by the Intramural Research Program of the NIH, National Institute of Environmental Health Sciences. Work in AS laboratory was supported by the UKRI Medical Research Council (MRC) grant MR/T028904/1. We are grateful to the NIEHS Collaborative crystallography group and the Advanced Photon Source (APS) SE Regional Collaborative Access Team (SER-CAT) 22-ID and 22-BM beam lines for assistance with crystallographic data collection. We appreciate our NIEHS colleague Dr. Chen Qiu for critical reading of this manuscript.

## Author contributions

**Guangning Zong**: Data curation; Formal analysis; Validation; Investigation; Visualization; Methodology; Writing—original draft; Writing—review and editing. **Yann Desfougères**: Data curation; Formal analysis; Investigation; Methodology. **Paloma Portela-Torres**: Data curation; Formal analysis; Investigation; Methodology. **Yong-Uk Kwon**: Resources. **Adolfo Saiardi**: Conceptualization; Resources; Data curation; Formal analysis; Supervision; Funding acquisition; Validation; Investigation; Visualization; Methodology; Writing—original draft; Writing—review and editing. **Stephen B Shears**: Conceptualization; Resources; Data curation; Formal analysis; Supervision; Funding acquisition; Validation; Investigation; Visualization; Methodology; Writing—original draft; Writing—review and editing. **Huanchen Wang**: Conceptualization; Resources; Data curation; Formal analysis; Supervision; Validation; Investigation; Visualization; Methodology; Writing—original draft; Writing—review and editing.

## Disclosure and competing interests statement

The authors declare no competing interests.

# Expanded View Figures

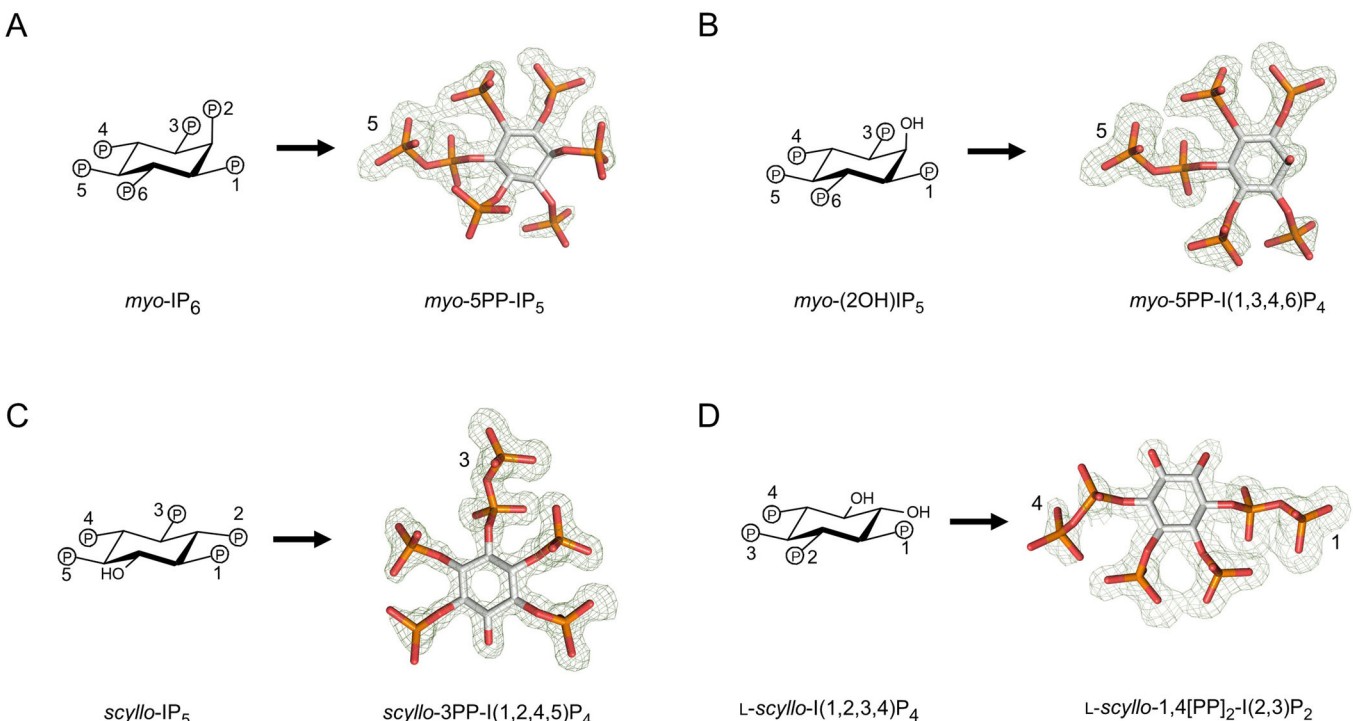

**Figure EV1.  Configurations of substrates captured in ADP/*Tv*IPK crystal complexes, and the structures of the corresponding products identified by their soaking into crystals of *Hs*DIPP1.**

(A–D) substrates are shown as chemical structures, and products are shown as stick models, in which carbon is gray, phosphorous is orange, and oxygen is red. The Fo-Fc electron density maps (green mesh) are contoured at 3σ. Substituent numbering follows standard nomenclature. Each structure is aligned so that the site of *Tv*IPK phosphorylation projects to the left, with the exception of the doubly phosphorylated L-*scyllo*-1,4[PP]$_2$-(2,3)IP$_2$.

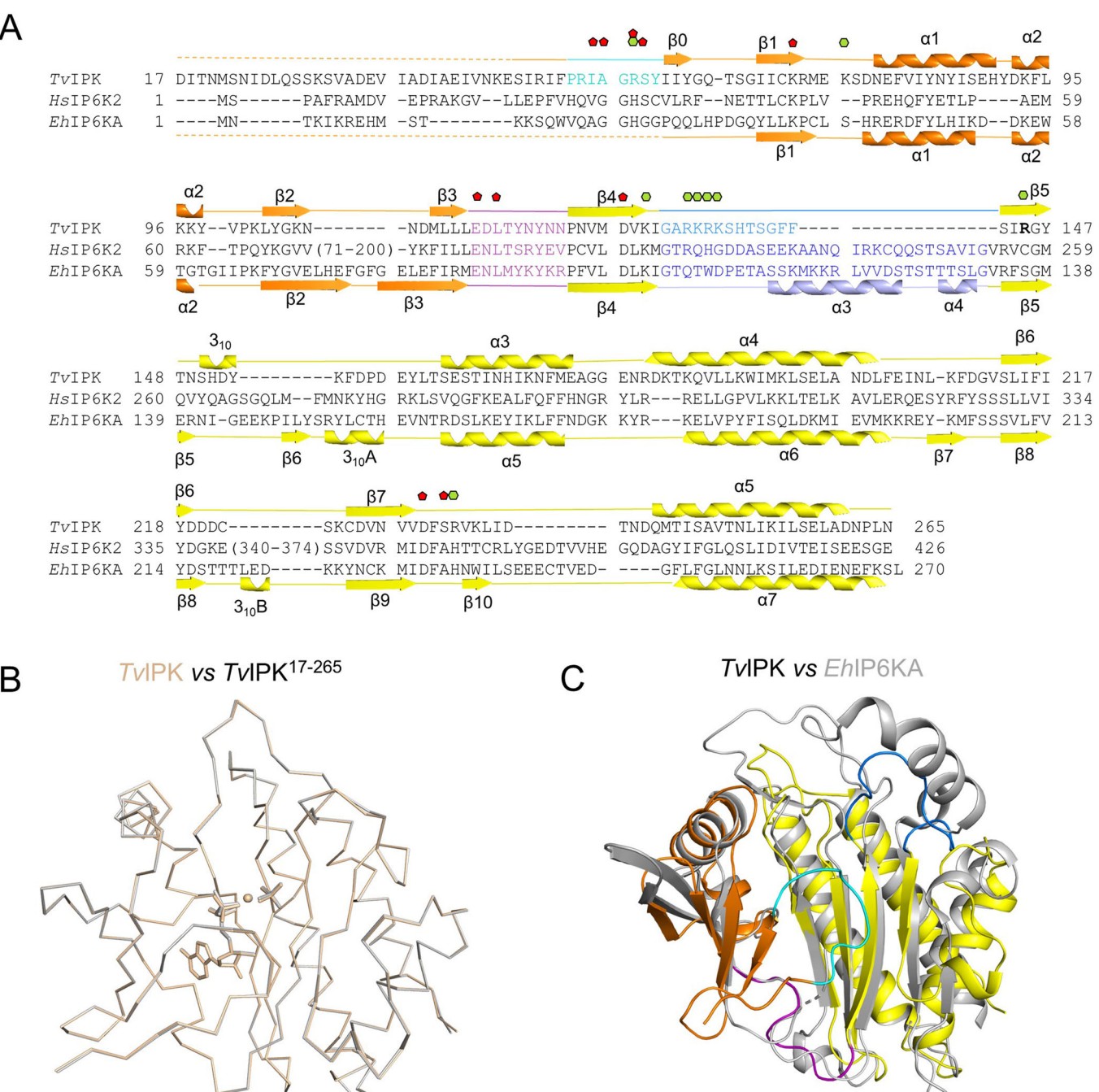

**Figure EV2. Structural comparisons of *Tv*IPK with IP6Ks.**

(**A**) A manually created, structure-based multiple sequence alignment of *Tv*IPK[17-265] with *Hs*IP6K2 (Accession number NP_057375.2; structural prediction by Alphafold) and *Eh*IP6KA (Accession number XP-648490.2; PBD: 4O4D). The known secondary structural elements of *Tv*IPK and *Eh*IP6KA are highlighted: α-helices and β-sheets are colored orange for their position in the N-lobe, and yellow for the C-lobe. The proposed G-loop in *Tv*IPK is colored cyan. Other known or predicted structural elements are colored as follows: hinge regions are colored purple, and the IP-binding elements are colored either light blue (for the purely loop structure in *Tv*IPK) or dark blue, for both *Hs*IP6K2 and *Eh*IP6KA. Residues that interact with substrates are highlighted with green hexagons; residues that form polar contacts with nucleotide are highlighted with red pentagons. (**B**) Superimposition of *Tv*IPK full length (colored wheat) and *Tv*IPK[17-265] (colored gray) in ribbon format; RMSD = 0.12 Å, 1589 comparable atoms. (**C**) Superimposition of *Tv*IPK (colored as Fig. 3B) and *Eh*IP6KA (colored as gray) in ribbon format. RMSD = 2.0 Å for 753 comparable atoms.

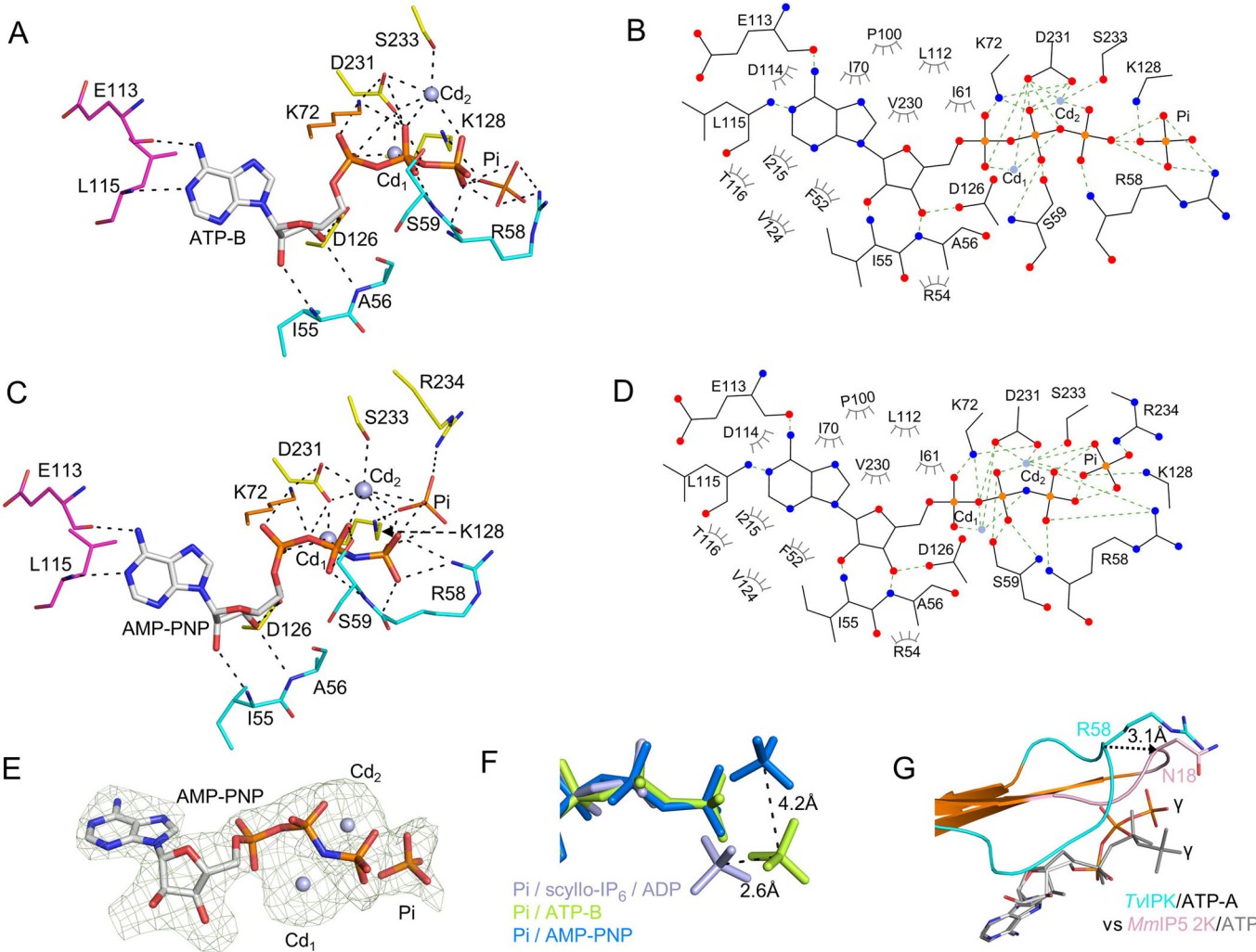

**Figure EV3. The nucleotide binding site.**

(A) Polar contacts (broken lines) of ATP in conformation B with surrounding residues, depicted in stick format. (B) Rendering by Ligplot+ of interactions between nucleotide and protein; Van der Waals contacts are shown as "eyelash" graphics. (C) Polar contacts (broken lines) of AMP-PNP with surrounding residues, depicted in stick format. (D) Rendering by Ligplot+ of interactions between nucleotide and protein; Van der Waals contacts are shown as "eyelash" graphics. (E) Omit electron density map of AMP-PNP, Cd and Pi. The Fo-Fc electron density map (green mesh) is contoured at 2.5 σ. (F) Superimposition of nucleotide phosphates and Pi from three *Tv*IPK structural complexes containing either Pi/scyllo-IP$_6$/ADP (light blue), Pi/ATP-B (green) or Pi/AMP-PNP (blue). (G) Stick model superimposition of the G-loops from *Tv*IPK (blue, in complex with ATP-A) and *Mm*IP5 2 K (green; in complex with ATP; PDB: 5MW8), represented as tubes that trace Cα atoms, except for the stick models of the indicated side chains.

 *The EMBO Journal* Volume 43 | February 2024 | 462 – 480 **EV3**

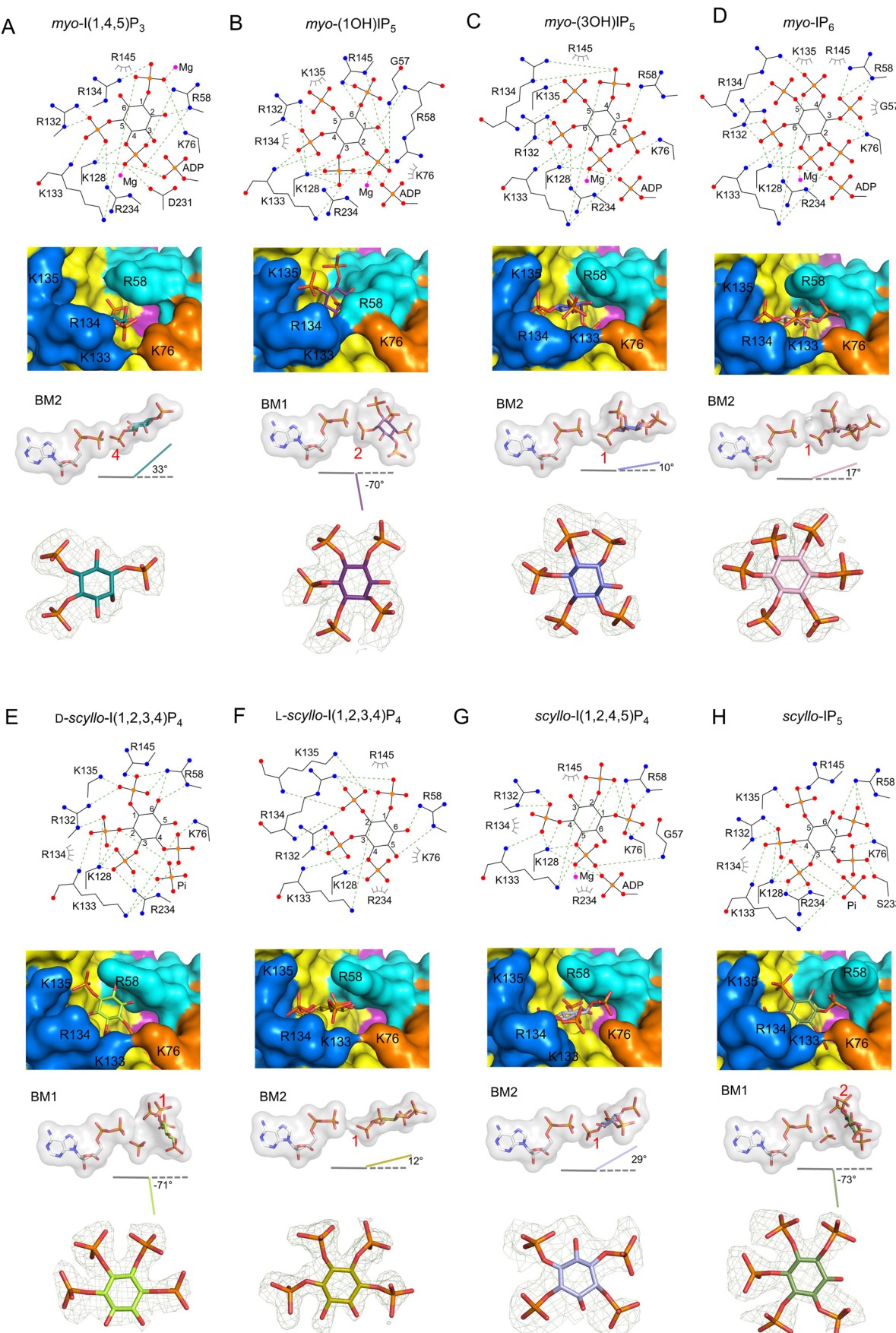

◀ **Figure EV4. The characteristics of substrate binding to *Tv*IPK.**

(A–H) Each vertical column from top to bottom depicts: a rendition created with Ligplot+ of substrate/protein interactions, *above* a space-filling depiction of the ligand binding pocket (color-coded as in Fig. 3A), above a space-filling representation of substrate and ADP (the IP phosphate closest to ADP is numbered). Note the graphical representations of the angle at which the plane of the inositol ring intersects a second plane that runs between the α- and β-phosphorus atoms of ADP through the bridging oxygen The angle of intersection between these two planes were divided into two groups (binding mode 1 and 2, i.e., BM1 and BM2). The angle for BM1 ranges from −73° to −70°); the angle for BM2 varies between +10° to +33°). The bottom panel of each column depicts the substrate's Fo-Fc electron density map (green mesh), contoured at 2.5 σ.

 

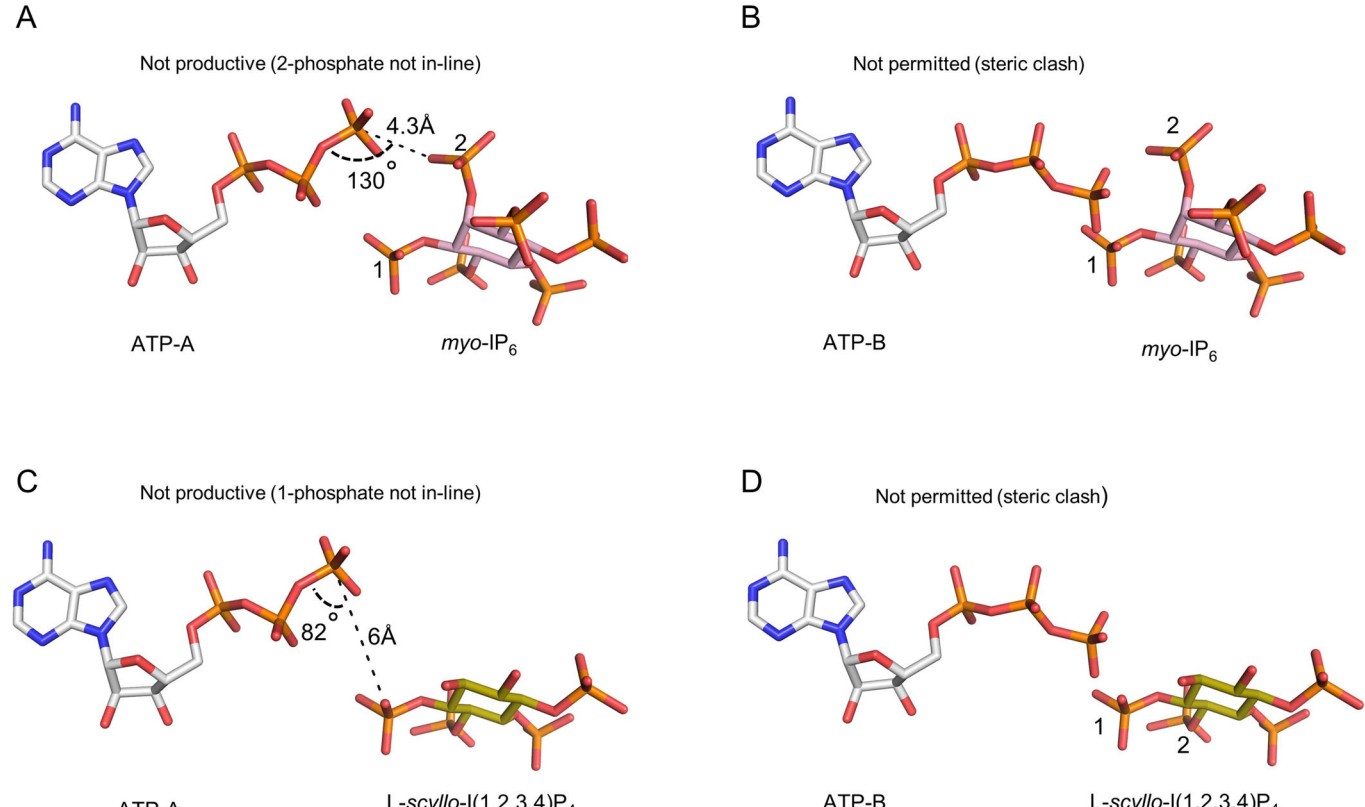

**Figure EV5. Superimpositions of ATP-A and ATP-B upon the orientations of *myo*-IP$_6$ and L-*scyllo*-I(1,2,3,4)P$_4$.**

(A–D) The configurations of ATP-A and ATP-B are taken from the *Tv*IPK/ATP complex described in Fig. 4. The configurations of *myo*-IP$_6$ and L-*scyllo*-I(1,2,3,4)P$_4$ are taken from the *Tv*IPK/ADP/substrate complexes described in Figure EV4D,F. For each of these superimpositions, the adenosine moiety has a consistent configuration and so was utilized as the reference point.

