## [Peer Review File · The EMBO Journal]

Biochemical and structural characterization of an inositol pyrophosphate kinase from a giant virus

Huanchen Wang, Stephen B. Shears, Adolfo Saiardi, Yong-Uk Kwon, Guangning Zong, Yann Desfougères, and Paloma Portela-Torres

DOI: [10.15252/emj.2023114992](https://doi.org/10.15252/emj.2023114992)

Corresponding authors: *Huanchen Wang* (huanchen.wang@nih.gov) , *Stephen B. Shears* (shears@niehs.nih.gov), *Adolfo Saiardi* (a.saiardi@ucl.ac.uk)

Review Timeline:

Submission Date:	13th Jul 23
Editorial Decision:	21st Aug 23
Revision Received:	19th Oct 23
Editorial Decision:	8th Nov 23
Revision Received:	11th Nov 23
Accepted:	15th Nov 23

Editor: Ieva Gailite

Transaction Report:

Dear Dr. Wang,

Thank you for submitting your manuscript for consideration by the EMBO Journal. We have now received comments from three reviewers, which are included below for your information.

As you will see from the reports, the reviewers appreciate the work, while also indicating a number of constructive points that would need to be addressed before acceptance here. From my side, I find the reviewer comments reasonable and constructive. Therefore, based on these positive assessments, I would like to invite you to address the issues raised by the reviewers in a revised manuscript. I would be happy to discuss the revision in more detail via email or phone/videoconferencing.

We generally allow three months as standard revision time. As a matter of policy, competing manuscripts published during this period will not negatively impact on our assessment of the conceptual advance presented by your study. However, please contact me as soon as possible upon publication of any related work to discuss the appropriate course of action. Should you foresee a problem in meeting this three-month deadline, please contact us to arrange an extension.

When preparing your letter of response to the referees' comments, please bear in mind that this will form part of the Review Process File and will therefore be available online to the community. For more details on our Transparent Editorial Process, please visit our website: <https://www.embopress.org/page/journal/14602075/authorguide#transparentprocess>. Please also see the attached instructions for further guidelines on preparation of the revised manuscript.

Please feel free to contact me if you have any further questions regarding the revision. Thank you for the opportunity to consider your work for publication, and I look forward to your revision.

With best regards,

leva

leva Gailite, PhD
Senior Scientific Editor
The EMBO Journal
Meyerohofstrasse 1
D-69117 Heidelberg
Tel: +4962218891309
i.gailite@embojournal.org

We realize that it is difficult to revise to a specific deadline. In the interest of protecting the conceptual advance provided by the work, we recommend a revision within 3 months (19th Nov 2023). Please discuss the revision progress ahead of this time with the editor if you require more time to complete the revisions.

Referee #1:

The manuscript by Zong et al reports the structural and enzymatic characterization of a inositol pyrophosphate kinase from a nucleocytoplasmic large DNA virus of the Mimiviridae family. Overall I find this a very detailed, well designed study of high technical quality. The (many) crystal structures have been refined with excellent statistics and geometry, and the enzyme and yeast assays are well described. In my view, the novelty of this work lies in the identification of a bona fide viral IPK with broad substrate specificity. My major criticism of this work is that as little is known about this virus, its potential host(s) and their respective InsP / PP-InsP metabolism, the implications of the findings are rather speculative. However, this is not a study design problem, but rather a direct consequence of the metagenome approach. I will summarize my comments and suggestions below:

0. General: I would suggest to remove statements of priority throughout the manuscript.

1. Major: Given that TvIPK sequences has been derived from a metagenome study (is an entire genome of Terrestriovirus available? ESTs?) one cannot be sure that TvIPK is actually an expressed gene and active enzyme in the presently unknown host. Since the entire study is based on a codon-optimized synthetic gene this potential limitation should, in my opinion, be clearly stated in either the introduction or discussion sections.

2. Minor: This statement from the introduction is not supported by a reference "Their biological sources are unknown, but it is plausible that amoeba scavenge them from soils by phagocytosis, so TvIPK could naturally encounter them." What is the evidence that amoeba are the hosts for these viruses? Along these lines, while the exact host for this virus is presently unknown, can the authors comment on the InsP/PP-InsP metabolism of other, characterized amoeba or in protozoans in general? The structure is later on compared to an enzyme from *Entamoeba histolytica*, does that imply that IP metabolism of this organism has been characterized?

3. Minor: While InsPs with three or more phosphates appear to be good substrates for TvIPK, is it formally possible that this kinases acts on different substrates in its host? Or does that charges nature of the substrate binding site exclude other possibilities?

4. Major: I understand from Figure 2 that TvIPK can phosphorylate pools of IP₃ to some PP-InsPs in yeast cells. But did you assay if it can phosphorylate IP₆ to 5-PP-IP₅, as it does in vitro? For example in a *ksc1/vip1* mutant? Is it possible to compare the protein expression levels of the synthetic TvIPK construct to endogenous yeast IPKs? In other word to what extend is the viral protein overexpressed in the yeast cells? The plasmid links to this paper ([https://www.jbc.org/article/S0021-9258\(17\)49846-2/fulltext](https://www.jbc.org/article/S0021-9258(17)49846-2/fulltext)), but I could not locate the promoter used.

5. Minor: R.m.s.d. values should be reported for the structural comparisons of TvIPK and EhIP6KA (pdb-id 4O4D) in Figure EV2. To me it seems that the two structures are highly similar, perhaps the result section "overall structure" could be shortened?

6. Minor: How do the authors interpret the ATP-B configuration? As a crystallization artifact? Is there one molecule per asymmetric unit and each protomer contains both configurations consistently across the different crystal structures? Does the ADP bound structure behave similarly?

7. Minor: Where does the Pi in the AMP-PNP complex originate from?

Referee #2:

The current work for the first time reports the presence of functional viral inositol phosphate kinase, TvIPK encoded by Terrestriovirus, a nucleocytoplasmic large ("giant") DNA virus (NCLDV). The overall quality of biochemical and structural data seem original and superb. The only minor concern is Figure 2C-F. I would strongly recommend similar gain-of-function experiments by using mammalian cells such as HEK293 cells. In addition, more full descriptions on the possible roles of this viral IPK in its life cycle as well as host cell physiology should be provided in the Discussion.

Referee #3:

The article unveils the discovery and thorough characterization of a novel enzyme, TvIPK, from a giant DNA virus (NCLDV), belonging to the inositol polyphosphate (InsP) kinase (IPK) family. The authors demonstrate that TvIPK can phosphorylate various scyllo- and myo-InsPs showing unambiguously that the activity is a pyrophosphorylation, i.e., the enzyme forms PP-InsPs. They show the activity both in vitro and in vivo assays, using HPLC; crystallography, mutagenesis, activity assays, yeast experiments etc. The authors have also determined the exact nature of the PP-InsP product formed in four instances by crystallizing them in complex with DIPP1 (a phosphatase that processes the PP-InsPs).

Also, the TvIPK structure has been characterized by X-ray crystallography, revealing, as expected, fold conservation with the IPK family. Some notable and specific features as a well-ordered G-loop for ATP binding and the shortest IP-loop inside the IPK family have been identified. The authors also produced multiple crystal complexes with ADP, ATP, or various InsPs providing a full characterization of the ligands binding site what allows determining two ATP binding forms (A and B) and two InsP binding modes (1 and 2). Although they have not obtained a productive binary or ternary complex that explained the nature of the products identified with DIPP1 crystal analysis, the structural information allowed them to rationalize how productive binding might be. Notably, obtaining multiple binding modes for InsP is a recurrent issue in this field due to their high symmetry. Until recently, IPKs were believed to be eukaryotic specific. The discovery of basic IPKs in viruses offers vital insights into the evolution of IP signaling. Particularly, as these enzymes are found in viruses that infect soil niches, rich in various inositol phosphates, they could have a significant impact on the environmental phosphorus cycle, particularly interesting due to permafrost thawing. This finding enhances understanding of both soil ecology and the ancestral development of InsP signaling in the primordial cell.

In my opinion, this work represents an obvious advancement in the field of inositides, particularly in expanding knowledge about PP-InsPs, which play crucial roles in energy and metabolism homeostasis in cells. The implications of this enzyme regarding ecological concerns in the medium to long term elevate this research's biological significance, likely appealing to a broad general interest. Technically, the research is well-developed, rationalizing multiple experiments to achieve a strictly deep characterization. I believe that this manuscript could be published in the designated journal after some minor issues are considered.

Following are minor concerns:

Comment 1

I wonder if, based on the HPLC results, it can be concluded that each substrate originates a unique PP-InsP isomer. For example, in figure 1B-E, PP-InsPs formed from 4 different substrates seem to elute in a similar time.

In the case that several PP-InsP isomers could be formed from each substrate, incubation with DIPP1 crystals could be selecting just one of them, with other products remaining unidentified.

If that is possible, could the crystallographic structures of TvIPK with the InsPs substrates be reflecting an InsP binding mode for the formation of a product different than the one identified and therefore productive modes for phosphorylation in different positions?

Maybe, in relation with this, the pictures could show the angle and distances between the reaction centers in the non-productive forms.

Comment 2

Regarding the nucleotide binding site, this site is described thoroughly and it is highly conserved within the IPK family. In particular, there is an extensive description for the G-loop, that certainly is an important element in PKs and IPKs. Other IPKs structures show disorder in this region, however structures for IP3KB and IP5 2K depicted also a well-ordered G-loop. Regarding the previously identified IPK G-loop interactions with the nucleotide, there are two sentences that in my opinion should be modified:

In pag 10: "G-loop interactions with the ATP phosphates have not previously been captured in structural analysis of members of the IPK family"

I believe that in IP3KB and in IP5 2K structures, the G-loop interacts with the ATP phosphates (codes 1aqx, 2xan, 5mw8).

Therefore, in IPKs, the interaction of the G-loop with the ATP phosphates has been captured previously. However, there are substantial differences from previous IPK structures that might be well described, specifically in comparison with IP3K and IP5 2K.

On the contrary, In Pag. 11 "The G-loop in MmIP3KB and IP5 2K has also been shown to interact with the 2' and 3' hydroxyl groups of the ribose moiety"

This statement appears ok for IP3KB, but I believe that not for IP5 2K. In IP5 2K the ribose OHs do not seem to interact with the G-loop.

Comment 3

I missed some data on how co-crystallization of DIPP1 with the TrIPK products has been performed.

Dear Dr. Gailite,

We appreciate the constructive comments made by the reviewers of our manuscript, which we have revised accordingly.

Referee 1

Major point (1). *Given that TvIPK sequences has been derived from a metagenome study (is an entire genome of Terrestriovirus available? ESTs?) one cannot be sure that TvIPK is actually an expressed gene and active enzyme in the presently unknown host. Since the entire study is based on a codon-optimized synthetic gene this potential limitation should, in my opinion, be clearly stated in either the introduction or discussion sections.*

We completely agree with this excellent point, which has been added to the Introduction (page 4, lines 1-8).

Minor point (2). *This statement from the introduction is not supported by a reference "Their biological sources are unknown, but it is plausible that amoeba scavenge them from soils by phagocytosis, so TvIPK could naturally encounter them." What is the evidence that amoeba are the hosts for these viruses? Along these lines, while the exact host for this virus is presently unknown, can the authors comment on the InsP/PP-InsP metabolism of other, characterized amoeba or in protozoans in general? The structure is later on compared to an enzyme from Entamoeba histolytica, does that imply that IP metabolism of this organism has been characterized?*

It is generally believed (e.g. see (Colson *et al*, 2017)) that amoeba are the natural hosts for mimivirus (which includes *Terrestriovirus*); a well-studied soil amoeba has been shown to host other Mimiviridae species (Alempic *et al*, 2023; Denet *et al*, 2017; Schulz *et al*, 2018). We now make this point on page 4, lines 3-5. Additionally, on page 5, last paragraph, we note that the *Terrestriovirus* genome was first identified in soils, an amoeba ecological niche.

On page 6, lines 4-5, we add a citation in support of amoeba accessing IPs from soils by phagocytosis.

As for adding a comment on IP metabolism of other, characterized amoeba, we have noted that this has been performed in most detail in the social amoeba, *Dictyostelium discoideum* (page 5, last paragraph). IP metabolism in *Entamoeba histolytica* is not well characterized, reference to this species is only made to its IP6 kinase, simply because it has proved a tractable model for enzymatic and structural studies (page 8, line 20).

Minor point (3) *While InsPs with three or more phosphates appear to be good substrates for TvIPK, is it formally possible that this kinases acts on different substrates in its host? Or does that charges nature of the substrate binding site exclude other possibilities?*

We do not formally exclude that other non-IP molecules could be metabolized by TvIPK. However, we consider this possibility too remote to be worth mentioning, since our thorough crystallographic data demonstrate that the catalytic core is precisely engineered to pyrophosphorylate IPs, as are the eukaryotic inositol polyphosphate kinases which in turn have never been shown to act on other substrates.

Major Point (4). *I understand from Figure 2 that TvIPK can phosphorylate pools of IP3 to some PP-InsPs in yeast cells. But did you assay if it can phosphorylate IP6 to 5-PP-IP5, as it does in vitro? For example in*

a ksc1/vip1 mutant? Is it possible to compare the protein expression levels of the synthetic TvIPK construct to endogenous yeast IPKs? In other word to what extent is the viral protein overexpressed in the yeast cells? The plasmid links to this paper ([https://www.jbc.org/article/S0021-9258\(17\)49846-2/fulltext](https://www.jbc.org/article/S0021-9258(17)49846-2/fulltext)), but I could not locate the promoter used.

Testing the ability of TvIPK to phosphorylate IP6 in vipΔ yeast is an excellent suggestion by the referee. Positive data were obtained and presented as new Figure 2F, as discussed on page 7, lines 25-26.

Unfortunately, it is not possible to compare levels of expression of TvIPK with those of yeast endogenous IP-kinases because the appropriate antibodies are not available.

We are sorry the promoter details were originally missing from our Methods Section. These have now been added to page 17, last paragraph.

Minor point (5). *R.m.s.d. values should be reported for the structural comparisons of TvIPK and EhIP6KA (pdb-id 4O4D) in Figure EV2. To me it seems that the two structures are highly similar, perhaps the result section "overall structure" could be shortened?*

This is another helpful suggestion. We added the RMSD value to the Figure EV2 legend, and we have considerably shortened this section of the manuscript (page 7-8).

Minor point (6) *How do the authors interpret the ATP-B configuration? As a crystallization artifact? Is there one molecule per asymmetric unit and each protomer contains both configurations consistently across the different crystal structures? Does the ADP bound structure behave similarly?*

Neither of the two ATP conformations are a crystallization artifact. The two alternative conformations (ATP-A and ATP-B) are output by the 'occupancy refinement' feature within the PHENIX algorithm.

There is one molecule of ATP in each asymmetric unit. The conformational information we provide is the average of all ATP molecules within the entire crystal: 55% of the ATP molecules in configuration A, and 45% in configuration B (page 9, paragraph 2).

ADP was analyzed in the same way and only one conformation was observed, unsurprisingly, because virtually all of the difference between ATP-A and ATP-B is due to the alternate conformations of their gamma phosphates.

Minor point (7). *Where does the Pi in the AMP-PNP complex originate from?*

We suggest that Pi in the TvIPK / ADP complex is derived from the crystallization buffer and is retained through our soaking protocols; this includes one such protocol in which AMP-PNP is substituted for ADP. See page 8 line 15-17 and page 9 line 19.

Referee 2, Minor concerns.

The current work for the first time reports the presence of functional viral inositol phosphate kinase, TvIPK encoded by Terrestrivirus, a nucleocytoplasmic large ("giant") DNA virus (NCLDV). The overall quality of biochemical and structural data seem original and superb. The only minor concern is Figure 2C-F. I would strongly recommend similar gain-of-function experiments by using mammalian cells such as HEK293 cells. In addition, more full descriptions on the possible roles of this viral IPK in its life cycle as well as host cell physiology should be provided in the Discussion.

With regards to the first suggestion to perform gain-of-function experiments in HEK293 cells, although the referee categorizes this as a minor point, in reality it would not be trivial to ensure such experiments would be quantitatively reliable; we would be obliged to create a viral vector hosting TvIPK to derive a HEK293 line in which the kinase is universally expressed.

In any case, we have instead derived proof of principle that the referee requests through gain-of-function experiments in an alternate model eukaryote, *S. cerevisiae*, in response to Major point 4 from Referee #1 (New Fig 2F). We respectfully suggest little additional knowledge will now be acquired by repeating these same experiments in a second eukaryotic model.

As for adding full descriptions on the possible roles of this viral IPK in its life cycle as well as host cell, in the paragraph that begins on page 14 last 2 lines, we hypothesize that viral IPK could help facilitate the hypermetabolic state of the host cells that is critical for viral biogenesis. But since we do not know the *nature* of the host cell, we are unable to offer solid proposals of any additional, specific impact upon host physiology.

Referee 3, Minor concerns**Comment 1**

I wonder if, based on the HPLC results, it can be concluded that each substrate originates a unique PP-InsP isomer. For example, in figure 1B-E, PP-InsPs formed from 4 different substrates seem to elute in a similar time.

In the case that several PP-InsP isomers could be formed from each substrate, incubation with DIPP1 crystals could be selecting just one of them, with other products remaining unidentified.

If that is possible, could the crystallographic structures of TvIPK with the InsPs substrates be reflecting an InsP binding mode for the formation of a product different than the one identified and therefore productive modes for phosphorylation in different positions?

Maybe, in relation with this, the pictures could show the angle and distances between the reaction centers in the non-productive forms.

These are great suggestions. We have added the angles and distances as requested (New versions of Fig 5 and EV5). We have also added to the manuscript the caveat that there might be other PP-IP products that did not soak into DIPP (page 6, lines 18-19).

Comment 2

Regarding the previously identified IPK G-loop interactions with the nucleotide, there are two sentences that in my opinion should be modified:

In page 10: "G-loop interactions with the ATP phosphates have not previously been captured in structural analysis of members of the IPK family"

I believe that in IP3KB and in IP5 2K structures, the G-loop interacts with the ATP phosphates (codes 1aax, 2xan, 5mw8). Therefore, in IPKs, the interaction of the G-loop with the ATP phosphates has been captured previously.

Thank you for making these points. We have eliminated the erroneous statement concerning G-loop interactions with members of the IPK family (page 10, paragraph 2).

However, there are substantial differences from previous IPK structures that might be well described, specifically in comparison with IP3K and IP5 2K.

We have expanded a comparative analysis of the G-loop in TvIPK, MmIP3KB and IP5 2K (page 10, paragraph 2, page 11 paragraphs 1 and 2; new figures 4G and EV3G). We apologize for not previously giving adequate credit to the published structures of IP5 2Ks.

On the contrary, In Pag. 11 "The G-loop in MmIP3KB and IP5 2K has also been shown to interact with the 2' and 3' hydroxyl groups of the ribose moiety"

This statement appears ok for IP3KB, but I believe that not for IP5 2K. In IP5 2K the ribose OHs do not seem to interact with the G-loop.

Again, we are grateful for the referee drawing attention to an error, which has been corrected (page 11, lines 8-9).

Comment 3

I missed some data on how co-crystallization of DIPP1 with the TvIPK products has been performed.

We obtained the complex DIPP1 with TvIPK products by soaking, not co-crystallization. We apologize for these methods being missing from our original submission; they have now been added: page 17 paragraph 3.

In conclusion, we hope you can appreciate that we have adequately accommodated the helpful suggestions by all 3 referees.

On behalf of all authors,

Hunachen Wang

Citations

Alempic JM, Lartigue A, Goncharov AE, Grosse G, Strauss J, Tikhonov AN, Fedorov AN, Poirot O, Legendre M, Santini S *et al* (2023) An Update on Eukaryotic Viruses Revived from Ancient Permafrost. *Viruses* 15

Colson P, La Scola B, Levasseur A, Caetano-Anollés G, Raoult D (2017) Mimivirus: leading the way in the discovery of giant viruses of amoebae. *Nature Reviews Microbiology* 15: 243-254

Denet E, Coupat-Goutaland B, Nazaret S, Pélandakis M, Favre-Bonté S (2017) Diversity of free-living amoebae in soils and their associated human opportunistic bacteria. *Parasitology Research* 116: 3151-3162

Schulz F, Alteio L, Goudeau D, Ryan EM, Yu FB, Malmstrom RR, Blanchard J, Woyke T (2018) Hidden diversity of soil giant viruses. *Nat Commun* 9: 4881

Dear Dr. Wang,

Thank you for submitting a revised version of your manuscript. Your study has now been seen by two of the original referees, who find that their previous concerns have been addressed and now recommend acceptance of the manuscript.

There now remain only a few editorial points that have to be addressed before I can extend acceptance of the manuscript:

1. Please limit the number of keywords to maximum five.
2. Please check that the funding information is correct and identical both in the manuscript and our online system. Please consider including MR/T028904/1 as a Grant Reference Number if appropriate.
3. Please upload the main and EV figures as individual production quality figure files in the .eps, .tif, or .jpg format (one file per figure).
4. Please move Main and EV figure legends after the References.
5. Our data editors have flagged the following issues in figure legends that need correcting:
 - Please define the error bars in the legend of figure 2c.
6. CRediT has replaced the traditional author contributions section because it offers a systematic, machine-readable author contributions format that allows for more effective research assessment. Please remove the Authors Contributions from the manuscript and use the free text boxes beneath each contributing author's name in our online submission system to add specific details on the author's contribution. More information is available in our guide to authors.
7. In the Data Availability section, please add the accession ID and resolvable links for the datasets. More information about the format of this section can be found here: <https://www.embopress.org/page/journal/14602075/authorguide#dataavailability>.
8. Please rename Table S1 into Table EV1 and update the callouts in the text accordingly. Subsequently, Appendix Table S2 should be renamed into Appendix Table S1 and added to the Appendix file.
9. In the Appendix, please add page numbers and a brief table of contents. Please update the nomenclature to Appendix Figure S1, etc. in the figure legends. Appendix figure legends should be removed from the manuscript text file and placed below the corresponding figures.
10. Please assemble source data into one folder per figure and upload as .zip files. For example, the Source data files for figure 1 need to be saved in a single folder and this needs to be zipped and then uploaded as "SD figure 1.zip" file.
11. Papers published in The EMBO Journal are accompanied online by a 'Synopsis' to enhance discoverability of the manuscript. It consists of A) a short (1-2 sentences) summary of the findings and their significance, B) 3-4 bullet points highlighting key results and C) a synopsis image that is 550x300-600 pixels large (width x height, jpeg or png format). You can either show a model or key data in the synopsis image. Please note that the image size is rather small and that text needs to be readable at the final size. Please send us this information together with the revised manuscript.

With best wishes,

Ieva

We realize that it is difficult to revise to a specific deadline. In the interest of protecting the conceptual advance provided by the work, we recommend a revision within 3 months (6th Feb 2024). Please discuss the revision progress ahead of this time with the editor if you require more time to complete the revisions.

Referee #1:

The authors have addressed all comments and concerns raised in my initial review, I particularly like that they decided to do the genetic rescue experiment of the yeast *vip1* mutant. I find the revised manuscript to be acceptable for publication in the EMBO J.

Referee #3:

All previously noted issues have been satisfactorily addressed by the authors. Therefore, in my assessment, no further changes are needed. In my opinion, the manuscript would be ready to be considered for publication.

The authors addressed the minor editorial issues.

Editor accepted the revised manuscript.